# Isolation and characterization of a pangolin-borne HKU4-related coronavirus that potentially infects human-DPP4-transgenic mice

Luo-Yuan Xia [1,6], Zhen-Fei Wang[1,2,6], Xiao-Ming Cui[3,4,6], Yuan-Guo Li[2,6], Run-Ze Ye [1,6], Dai-Yun Zhu [3], Fang-Xu Li[2], Jie Zhang[3], Wen-Hao Wang[2], Ming-Zhu Zhang[1], Wan-Ying Gao[1], Lian-Feng Li[1], Teng-Cheng Que[5], Tie-Cheng Wang[2], Na Jia [3,4], Jia-Fu Jiang [3,4] ✉, Yu-Wei Gao[2] ✉ & Wu-Chun Cao [1,3,4] ✉

We recently detected a HKU4-related coronavirus in subgenus *Merbecovirus* (named pangolin-CoV-HKU4-P251T) from a Malayan pangolin[1]. Here we report isolation and characterization of pangolin-CoV-HKU4-P251T, the genome sequence of which is closest to that of a coronavirus from the greater bamboo bat (*Tylonycteris robustula*) in Yunnan Province, China, with a 94.3% nucleotide identity. Pangolin-CoV-HKU4-P251T is able to infect human cell lines, and replicates more efficiently in cells that express human-dipeptidyl-peptidase-4 (hDPP4)-expressing and pangolin-DPP4-expressing cells than in bat-DPP4-expressing cells. After intranasal inoculation with pangolin-CoV-HKU4-P251, hDPP4-transgenic female mice are likely infected, showing persistent viral RNA copy numbers in the lungs. Progressive interstitial pneumonia developed in the infected mice, characterized by the accumulation of macrophages, and increase of antiviral cytokines, proinflammatory cytokines, and chemokines in lung tissues. These findings suggest that the pangolin-borne HKU4-related coronavirus has a potential for emerging as a human pathogen by using hDPP4.

Entering the twenty-first century, sudden emergence and global epidemic of three highly pathogenic human coronaviruses, including SARS-CoV and SARS-CoV-2 in subgenus *Sarbecovirus* and MERS-CoV in subgenus *Merbecovirus*, have provoked the widespread worry about zoonotic coronaviruses, and raised the urgent need for identifying animal-originated coronaviruses with human-infecting potential.

Coronaviruses in pangolins were largely overlooked until the recent discovery of SARS-CoV-2-related viruses in Malayan pangolins (*Manis javanica*)[2–6]. Subsequently, SARS-CoV-2-related sarbecoviruses were reported in captive Malayan pangolins[7] as well as confiscated Malayan and Chinese pangolins (*M. pentadactyla*)[8–10]. SARS-CoV-2 neutralizing antibodies were detected in a pangolin from a wildlife checkpoint in

[1]Institute of EcoHealth, School of Public Health, Cheeloo College of Medicine, Shandong University, Jinan 250012 Shandong, P. R. China. [2]Changchun Veterinary Research Institute, Changchun, 130122 Jilin, P. R. China. [3]State Key Laboratory of Pathogen and Biosecurity, Beijing Institute of Microbiology and Epidemiology, Beijing 100071, P. R. China. [4]Research Unit of Discovery and Tracing of Natural Focus Diseases, Chinese Academy of Medical Sciences, Beijing 100071, P. R. China. [5]Terrestrial Wildlife Rescue and Epidemic Diseases Surveillance Center of Guangxi, Nanning, Guangxi, P. R. China. [6]These authors contributed equally: Luo-Yuan Xia, Zhen-Fei Wang, Xiao-Ming Cui, Yuan-Guo Li, Run-Ze Ye. ✉e-mail: jiangjf2008@139.com; gaoyuwei@gmail.com; caowuchun@126.com

Southern Thailand[11]. Besides the sarbecoviruses harbored in pangolins, we identified a virus in subgenus *Merbecovirus* from a trafficked Malayan pangolin through the meta-transcriptomic sequencing named pangolin-CoV-HKU4-P251T[1]. Another paper also reported a genetically close MjHKU4r-CoV from the pangolin anal swab samples[12].

In spite of the potential for cross-species transmission of pangolin-CoV-HKU4-P251T according to the in silico analysis, its virological characteristics and risk for emerging as a human pathogen are unknown due to the lack of an isolated virus. Considering that its closely related *Tylonycteris*-bat-CoV-HKU4 can utilize the human dipeptidyl-peptidase-4 (hDPP4)[13–16], which is the MERS-CoV receptor[17], we speculate that pangolin-CoV-HKU4-P251T might also use hDPP4. In terms of host, hDPP has higher identity with pangolin DPP (pDPP4) (89.0–89.3%) than with bat DPP4 (bDPP4) (82.4–83.2%), suggesting a risk for a more likely spillover event of pangolin-CoV-HKU4-P251T through using hDPP4[1]. Otherwise, a recent study found that MERS-CoV-related viruses from bats can use angiotensin-converting enzyme 2 (ACE2) as an entry receptor[18]. Therefore, it is essential to validate the infectivity of the pangolin-borne HKU4-related coronavirus through in vitro and in vivo experiments. Furthermore, its pathogenicity should also be evaluated through the animal infection.

In this work, we obtain the pangolin-CoV-HKU4-P251T isolate and find the virus can infect human cell lines through in vitro experiments. In addition, pangolin-CoV-HKU4-P251T is able to replicate efficiently in cells expressing human- and pangolin-dipeptidyl-peptidase-4 (DPP4).

Human DPP4 transgenic female mice may be infected by pangolin-CoV-HKU4-P251T via the respiratory pathway resulting in interstitial pneumonia. The pangolin-borne HKU4-related coronavirus may utilize the hDPP4 receptor to infect humans, causing a threat to human health.

## Results

### Isolation and characterization of pangolin-CoV-HKU4-P251T

We homogenized the sample, from which the whole genome sequence of pangolin-CoV-HKU4-P251T had been obtained[1], and then inoculated the supernatant into African green monkey kidney Vero 81 cells (ATCC, Cat. No. CCL-81) to isolate the virus. To prove the growth of the virus and quantify the viral loads, we conducted a quantitative RT-PCR (qRT-PCR) assay using specific primers for pangolin-CoV-HKU4-P251T (Supplementary Table 1). The viral copies kept increasing from $10^{7.1}$ copies/μL in the first passage to a maximum of over $10^{9.0}$ copies/μL in the thirteenth passage with a 75.9-fold rise, and maintained stationary afterward (Fig. 1a), showing the gradual adaptation of the virus to Vero 81 cells. The electron microscopy revealed that the viral particles in the supernatant of cell culture in the fifth passages displayed typical coronavirus morphology of 90–120 nm in diameter with clear spikes around the virus surface (Fig. 1b). To confirmed the replication of the virus in the cells, we performed fluorescence in situ hybridization (FISH) assays using specific probes for pangolin-CoV-HKU4-P251T (Supplementary Table 2), and visualized the viral signals in infected cells at 48 h post inoculation (HPI) (Fig. 1c).

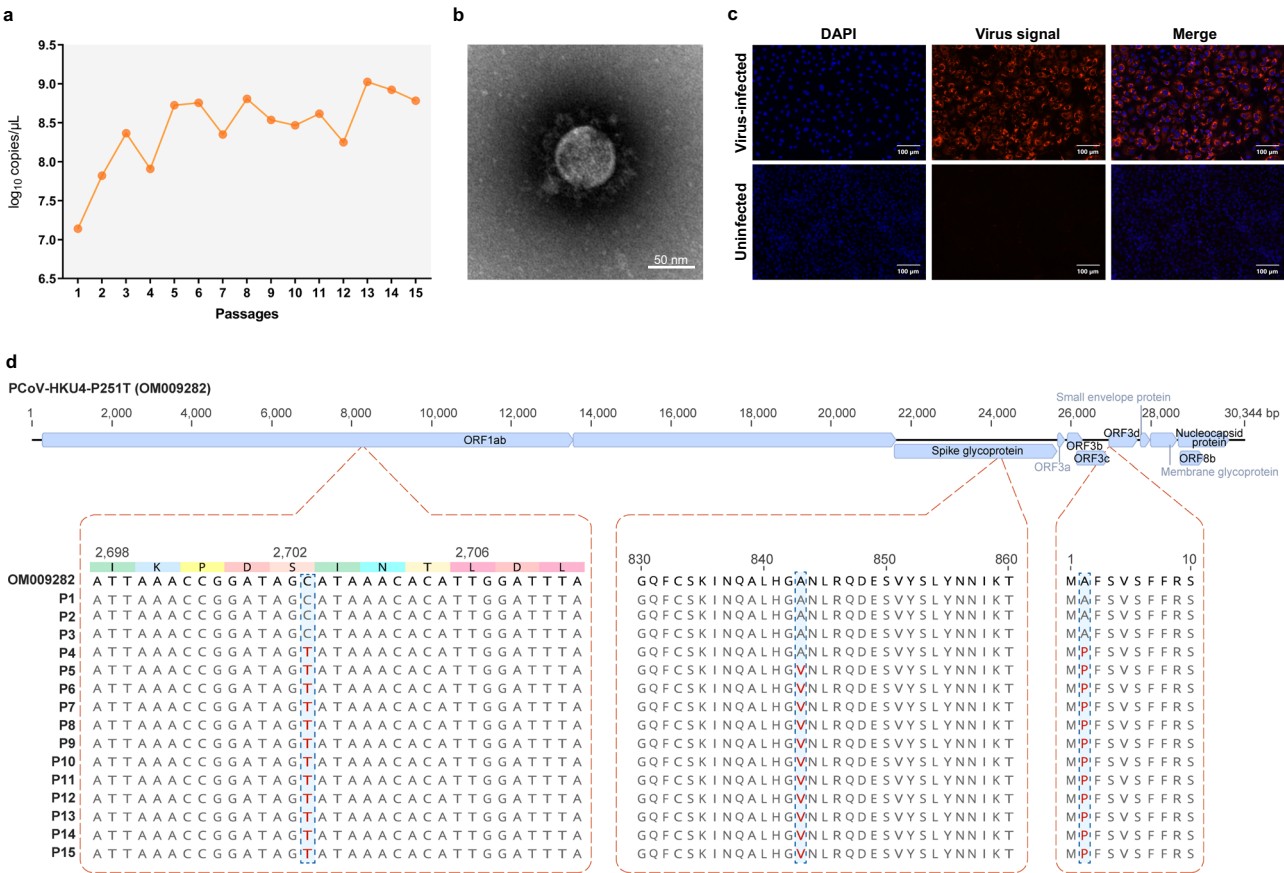

**Fig. 1 | The isolation and characteristics of pangolin-CoV-HKU4-P251T. a** The viral load of pangolin-CoV-HKU4-P251T in series passages in Vero 81 cell cultures. The viral load are expressed as copies per microliter (μL). **b** Negative stain electron microscopy verified the extracellular virus particle of pangolin-CoV-HKU4-P251T. Original magnification ×80 K. **c** Fluorescence in situ hybridization locating ORF1ab gene of pangolin-CoV-HKU4-P251T at 48 HPI. Nuclei, DAPI (blue); ORF1ab probe, Quasar 570 (red). Representative microscopy fields are shown. Original magnification ×200. Each imaging experiment was performed at least three times independently with similar results, and the representative microscopy fields are shown (**b** and **c**). **d** Genetic variation analysis of pangolin-CoV-HKU4-P251T during in vitro passage. The parental pangolin-CoV-HKU4-P251T strain (GenBank accession No. OM009282) was used as the reference sequence. Source data are provided as a Source Data file.

The genome sequence of pangolin-CoV-HKU4-P251T isolate in the first passage proved that the culture was pure, and corresponded to the sequence directly obtained from the raw sample through meta-transcriptomic sequencing. Our pangolin-CoV-HKU4-P251T genome (GenBank accession No. OM009282.1) was 30,344 bp in length, and shared 99.1% nucleotide (nt) identity with the genome sequence of MjHKU4r-CoV-1 from a Malayan pangolin (GenBank accession No. OQ786861.1)[12], and 94.3% nt identity with a coronavirus from the greater bamboo bat (*Tylonycteris robustula*) at Yunnan Province, China (GenBank accession No. ON745165.1), which was released in GenBank on Aug. 15, 2022 (Supplementary Fig. 1a). The amino acid (aa) identity in RNA-dependent RNA polymerase (RdRp) was 99.8% with MjHKU4r-CoV-1 and 98.7% with the coronavirus from a greater bamboo bat, which was higher than the previously reported 96.7% between pangolin-CoV-HKU4-P251T and *Tylonycteris*-bat-CoV-HKU4 from lesser bamboo bats at Hong Kong, China[1]. The spike protein of pangolin-CoV-HKU4-P251T showed 99.3% aa identity with MjHKU4r-CoV-1, and 91.8% with the coronavirus from the greater bamboo bat. The current findings indicate that pangolin-CoV-HKU4-P251T together with MjHKU4r-CoV-1 are genetically closest to the coronavirus from greater rather than lesser bamboo bats. Recombination is a primary mechanism for interspecies transmission and the emergence of novel strains[19], which has been observed in pangolin-borne sarbecoviruses[2]. Furthermore, a cross-family recombinant from coronavirus and other RNA viruses has been reported in bats[20]. Therefore, we did a recombination analysis, and found no recombinant signal in pangolin-CoV-HKU4-P251T (Supplementary Fig. 1b). Phylogenetic trees of different genomic regions revealed that pangolin-CoV-HKU4-P251T always clustered with the coronavirus from the greater bamboo bat and *Tylonycteris*-bat-CoV-HKU4 from lesser bamboo bats (Supplementary Fig. 1c), further proving no recombination occurrence.

We conducted viral sequencing of the 15 passages to assess possible mutation of pangolin-CoV-HKU4-P251T through subcultures (Supplementary Table 3), and found that it had a persistent synonymous substitution at position 2702 (AGC:AGT) of ORF1ab and a consistent non-synonymous substitution of A2P in NS3d protein from the fourth to 15th passage. A non-synonymous substitution leading to A843V in spike protein occurred since the fifth passage (Fig. 1d). These findings indicate the virus was rather stable during serial passages in vitro. Considering that pangolin-CoV-HKU4-P251T maintained stability during serial passage with only two non-synonymous mutations in the genome, the 15th passage viruses were used in the subsequent experiments due to the availability of a high-titer stock.

## Infectivity of pangolin-CoV-HKU4-P251T to human cells
To understand the cross-species transmission potential of pangolin-CoV-HKU4-P251T, we examined its infectivity to a variety of immortalized cell lines derived from human. Pangolin-CoV-HKU4-P251T was able to efficiently replicated in human cell lines derived from hepatoma (Huh7), colorectal adenocarcinoma (Caco-2), lung adenocarcinoma (Calu-3) and bronchial epithelioid (BEAS-2B), and kept continuous growth during the culturation, with 120.6, 202.4, 38.5 and 3.4-fold increase of viral copies in supernatants at 72 HPI, respectively (Fig. 2a). We conducted FISH using specific probes, and the viral signals were visualized in Huh7, Caco-2, Calu-3 and BEAS-2B cells (Fig. 2b), which confirmed the infectivity of pangolin-CoV-HKU4-P251T in multiple human cells. These findings suggest a human-infecting ability of pangolin-CoV-HKU4-P251T. The virus was unable to replicate in cell line derived from human bronchial carcinoma (ChaGo-K-1), and similarly no viral signal was observed in the cells when tested by FISH assay (Supplementary Fig. 2). To investigate if the differences in viral replication were possibly related to the expression of hDPP4 and hACE2 receptors among the cell lines, we conducted the western blotting assay. The hDPP4 expression was observed in all four cell lines susceptible to pangolin-CoV-HKU4-P251T. While hACE2 was expressed in

Vero 81, Caco-2 and Calu-3 cells, but not in Huh7 cell line, on which the virus most efficiently replicated (Fig. 2c). These findings suggest that the potential spill-over transmission of the virus might depend on hDPP4 rather than hACE2. Notably, obvious cytopathic effect (CPE) was visualized in the pangolin-CoV-HKU4-P251T-infected Huh7 cells (Fig. 2d), while CPE was subtle in either infected Vero 81 or Caco-2 and Calu-3 cells. Therefore, we used Huh7 cell line to quantify the viral growth and activity using the method of 50% tissue culture infective dose ($TCID_{50}$). The live virus titer was continuously increased from 24 to 72 HPI (Fig. 2e), indicating the persistent growth in the culture. Furthermore, we compared the kinetics of virus replication between the original isolate (passage 1) and in vitro adapted strain (passage 15) that used for the experiments in this study. The adapted viruses grew more efficiently in the initial stage reaching peak at 48 HPI, and slowed down thereafter. While the titer of the originally isolated virus was gradually increased and up to the comparable level with the adapted virus at 144 HPI (Fig. 2f).

Considering that pangolin-CoV-HKU4-P251T was genetically close to the coronavirus from greater bamboo bats as well as *Tylonycteris*-bat-CoV-HKU4 from lesser bamboo bats, we therefore tested bat lung cell line (Tb 1 Lu) for susceptibility to pangolin-CoV-HKU4-P251T. Notably, pangolin-CoV-HKU4-P251T did not replicate in Tb 1 Lu cells derived from a Brazilian free-tailed bat (*Tadarida brasiliensis*) (Supplementary Fig. 2).

## DPP4 utility of pangolin-CoV-HKU4-P251T In Vitro
To further investigate whether pangolin-CoV-HKU4-P251T can utilize DPP4 and ACE2 as receptors for infection, we conducted the experiments by observing the virus infectivity in wild type HeLa (HeLa-WT) cells as well as human DPP4-expressing HeLa (HeLa-hDPP4), *Manis javanica* pangolin DPP4-expressing HeLa (HeLa-pDPP4) and *T. pachypus* bat DPP4-expressing HeLa (HeLa-bDPP4), human ACE2-expressing HeLa (HeLa-hACE2), pangolin ACE2-expressing HeLa (HeLa-pACE2) and bat ACE2-expressing HeLa (HeLa-bACE2) cells (Fig. 3a, c). Pangolin-CoV-HKU4-P251T grew well and replicated efficiently in hDPP4-HeLa and pDPP4-HeLa cells, with 21.8 and 64.5-fold increase in viral copies, respectively, in comparison to viral copies in HeLa-WT cells at 48 HPI. The fold increase in viral copies was 5.0 in HeLa-bDPP4 cells, which was significantly lower than those in HeLa-hDPP4 and HeLa-pDPP4 cells (Fig. 3b). In contrast, ACE2-expressing cells did not show any increase in viral load compared to HeLa-WT cells, suggesting that ACE2 may not be an essential receptor for pangolin-CoV-HKU4-P251T (Fig. 3d).

We then examined the pangolin-CoV-HKU4-P251T-infected cells using indirect immunofluorescence assay (IFA), and observed obvious viral fluorescence signals in HeLa-hDPP4 and HeLa-pDPP4 cells, and only weaker fluorescence signals in HeLa-bDPP4 cells (Fig. 3e). No viral signals were observed in any kinds of ACE2-expressing cells (Fig. 3f). These findings indicate that pangolin-CoV-HKU4-P251T can efficiently utilize both hDPP4 and pDPP4 receptors rather than bDPP4 receptor to infect cells, further implying its adaptation to pangolins possibly as the reservoir hosts and potential infectivity to humans by using DPP4 rather than ACE2 receptors. The inefficient bDPP4 utility of pangolin-CoV-HKU4-P251T was in accordance with its inability to replicate in bat-derived Tb 1 Lu cells, further study is deserved to understand the evolutionary relationships between pangolin-CoV-HKU4-P251T and its close-related bat-CoV-HKU4 in either greater or lesser bamboo bats.

## Infectivity of pangolin-CoV-HKU4-P251T in hDPP4-transgenic mice
We evaluated the infectivity of pangolin-CoV-HKU4-P251T in hDPP4-transgenic C57BL/6 mice (hDPP4-mice) and wild type C57BL/6 mice (WT-mice) by intranasal inoculation. Considering the immature mice are usually used for evaluating the susceptibility of viruses in the subgenus *Merbecovirus* as reported by previous studies[12,16], 5-week-old specific pathogen-free female hDPP4-mice and WT-mice were

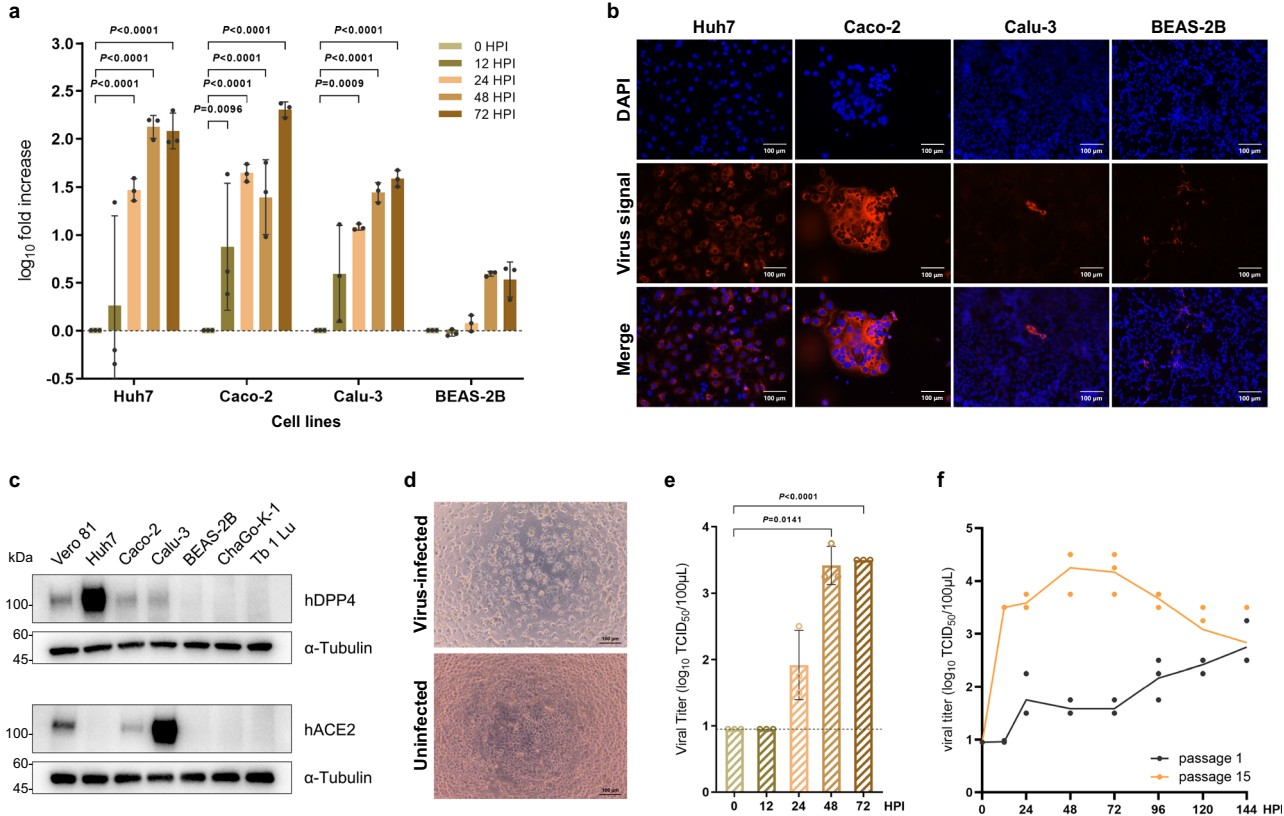

**Fig. 2 | Efficient replication of pangolin-CoV-HKU4-P251T in human cell lines.** **a** Kinetics of replication of pangolin-CoV-HKU4-P251T in human Huh7, Caco-2, Calu-3, and BEAS-2B cell lines. Viral RNA levels of virus in cell culture supernatant were detected at 0, 12, 24, 48, and 72 HPI, with three independent biological replicates per time point and three technical replicates per sample. The viral RNA levels were normalized relative to initial phase of infection (0 HPI). Data are presented as mean ± SD (shown as error bars). ANOVA was used for multiple comparisons. **b** Fluorescence in situ hybridization locating ORF1ab gene of Pangolin-CoV-HKU4-P251T in each cell lines at 48 HPI. Nuclei, DAPI (blue); ORF1ab probe, Quasar 570 (red). Representative microscopy fields are shown. Original magnification ×200. See also Supplementary Fig. 2. **c** Western blotting analyses of expressions of hDPP4 and hACE2 in various cell lines. A-Tubulin was used as the

loading control. **d** Cytopathic effect (CPE) of virus-infected Huh7 cells at 5 DPI. Original magnification×100. The uninfected Huh7 cells cultivated in parallel are used as controls. Each imaging experiment was independently performed at least three times with similar results, and representative images are shown (**b**–**d**). **e** Viral titers of infected Huh7 cells at different time points post-infection. $n = 3$ biologically independent experiments per time point. Data are presented as mean ± SD (shown as error bars). ANOVA was used for multiple comparisons. **f** The growth curve of passages 1 and 15 viruses in Huh7 cells. Cells were infected with the virus at an MOI of 0.01. At the specified time intervals, the supernatants were collected and the viral titers were measured as instructed. $n = 3$ biologically independent experiments per time point. Data are displayed as a line representing the mean, with individual data points shown as dots. Source data are provided as a Source Data file.

inoculated intranasally with pangolin-CoV-HKU4-P251T at a volume of 50 μL virus stock with a viral titer of $10^4$ TCID$_{50}$. The control groups received an equivalent volume of cell culture supernatant or heat-inactivated virus. Three mice per group were necropsied after euthanasia at each time point, from which nasal turbinate, lung, heart, brain, kidney, spleen, liver, and small intestine samples were collected to examine hDPP4 expression, viral replication, and pathological changes (Supplementary Fig. 3a).

The western blotting analysis revealed hDPP4 expression in all tissues collected from hDPP4-transgenic mice, while no expression was observed in any tissues from wild-type mice (Supplementary Fig. 3b). Pangolin-CoV-HKU4-P251T RNA was detected by qRT-PCR in lungs of infected hDPP4-mice at 1, 3, 6, 9 and 12 DPI (Fig. 4a). Although the viral load in lung at 1 DPI ($10^{9.8}$ copies/g, 95% CI: $10^{9.4}$–$10^{10.0}$ copies/g) was significantly higher than those at other timepoints ($P < 0.001$), the difference in viral copies was not significant from 3 to 12 DPI ($P = 0.135$). The viral RNA was not detectable in any other tissues of hDPP4-mice infected with pangolin-CoV-HKU4-P251T at any timepoint (Supplementary Fig. 3c). No viral RNA was tested either in hDPP4-mice in the control groups inoculated with cell culture supernatant and heat-inactivated virus, or in any WT-mice. To verify the presence of the replication of the virus, we examined the relative expression of

subgenomic mRNA (sgmRNA) in the lung samples. The sgmRNA of the virus was detected only on the first day, and the expression was significantly higher in hDPP4-mice than in WT-mice (Fig. 4b). To evaluate the seroconversion of virus-infected mice, serum was collected from all mice, and the serum neutralizing antibody assay against pangolin-CoV-HKU4-P251T was conducted. Neutralizing antibodies were developed in one-third of virus-infected hDPP4-mice, whereas no neutralizing antibody was detected in the serum of the virus-infected WT-mice and any control mice (Supplementary Table 4).

To prove presence of infectious viruses in the tissues positive for qRT-PCR, we conducted immunohistochemical (IHC) staining test in tissue sections with the specific antibodies against viral nucleocapsid proteins of pangolin-CoV-HKU4-P251T customized in Sino Biological Inc. Viral antigens were found in alveolar epithelia and degenerative bronchial epithelial cells, as well as various inflammatory cells of the lung tissues from the infected hDPP4-mice, when we observed using samples collected at 3, 6, 12 DPI (Fig. 4c). The IHC staining results was consistent with those of qRT-PCR tests, and confirmed pangolin-CoV-HKU4-P251T infectivity in lungs of hDPP4-mice. Furthermore, pangolin-CoV-HKU4-P251T was respectively isolated from the positive lung tissues of hDPP4-mice. The viral isolates from these tissues were confirmed by viral genome sequencing. There was no meaningful

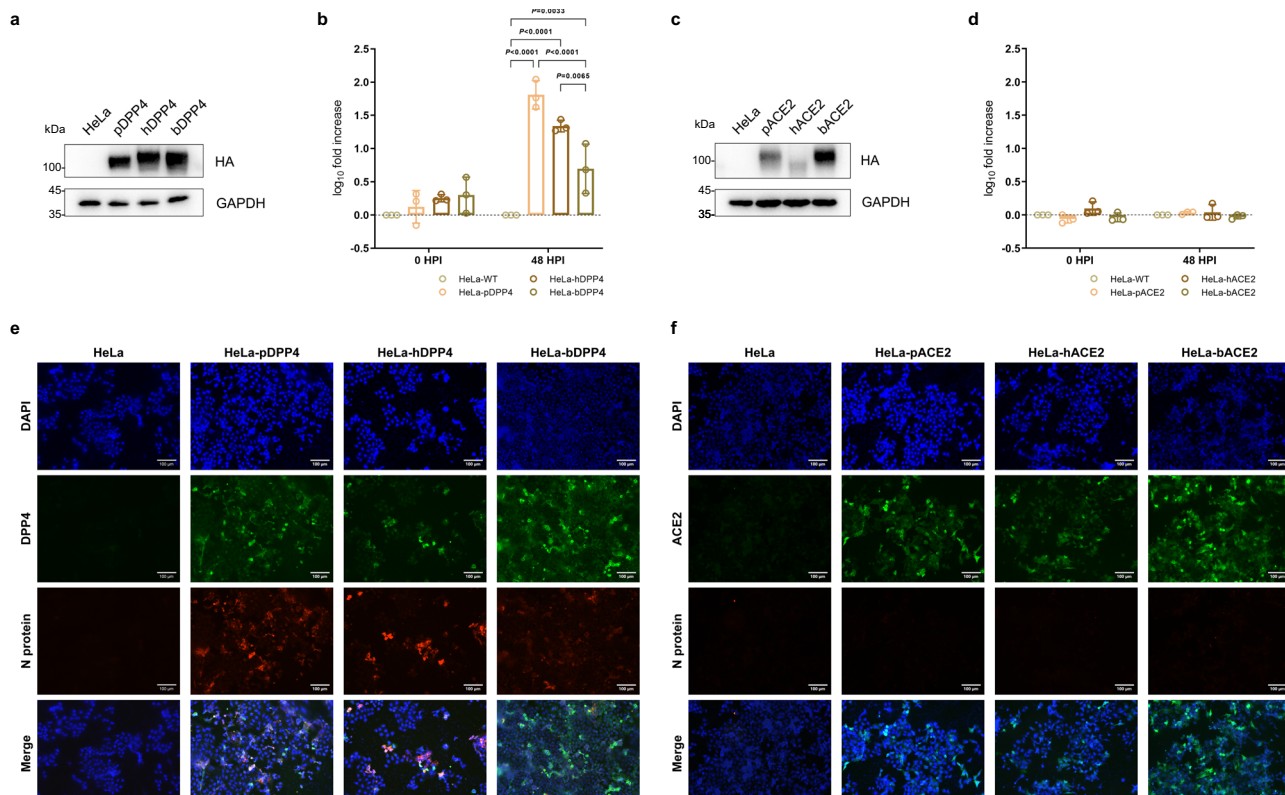

**Fig. 3 | DPP4 usage of pangolin-CoV-HKU4-P251T In Vitro.** Western blotting assay showing the expression of DPP4 (**a**) and ACE2 (**c**) in transient transfected cells. GAPDH was used as the loading control. Pangolin-CoV-HKU4-P251T growth in HeLa cells overexpression DPP4 (**b**) and ACE2 (**d**). The HeLa-WT cells were used as a negative control. Fold increases of viral copies are determined by comparing to the viral copies in HeLa-WT cells. Data are presented as mean ± SD (shown as error bars) of three independent biological replicates per time point and three technical replicates per sample. ANOVA was used for multiple comparisons. **e**, **f** Indirect immunofluorescence staining. Representative images show the expression of receptor protein (green) and viral nucleocapsid protein (red) in different cell lines at 48 HPI. Nuclei, DAPI (blue). Original magnification ×200. Each imaging experiment was performed independently at least three times with similar results, and representative images of the results are shown (**a**, **c**, **e**, and **f**). Source data are provided as a Source Data file.

non-synonymous mutation associated with virus host-adaptability and pathogenicity identified through the genome sequence analysis of the infected lungs collected at different times after infection (Supplementary Fig. 3d).

## Pathogenicity of pangolin-CoV-HKU4-P251T in hDPP4-transgenic mice

After inoculation, mice were weighed and observed daily for clinical signs of disease. No death or distinctive clinical sign was observed in each group throughout the experiment. The percentages of body weight change for each group were estimated by comparing the group average weight with their initial average weight. The body weights of hDPP4-mice in the control group maintained about 10% gain since 1 DPI. While the weights of pangolin-CoV-HKU4-P251T-infected hDPP4-mice always lower than those of control groups inoculated with either cell culture supernatant or heat-inactivated virus during the experiment period, and was decreased since 5 DPI (all $P < 0.05$), with a weight loss of up to 4.6% at 9 DPI (Fig. 5a). The body weights of WT-mice infected with pangolin-CoV-HKU4-P251T kept comparable to those of uninfected control WT-mice. Although no obvious clinical sign was observed, the hDPP4-mice infected with pangolin-CoV-HKU4-P251T did not gain weight in contrast to either the uninfected hDPP4-mice or wild-type mice, and even showed body weight loss in the late stage of infection, suggesting that hDPP4-mice had been affected by the virus infection.

We then conducted histopathological assay to observe microscopic lesions in lungs of the infected hDPP4-mice. The right lobe of each infected mouse was fixed, sectioned, and stained with hematoxylin and eosin (H&E). A lung sample from one uninfected hDPP4-mouse was used as a negative control. H&E staining revealed that pangolin-CoV-HKU4-P251T-infected hDPP4-mice at 3 DPI developed interstitial pneumonia, characterized by minor multifocal lesions and infiltration of inflammatory cells. Progressive damages in lungs of pangolin-CoV-HKU4-P251T-infected hDPP4-mice were visualized. At 6 and 12 DPI, obvious interstitial pneumonia with diffuse lesions characterized by thickened alveolar septa, multifocal alveolar hemorrhage and accumulation of inflammatory cells in more alveolar cavities. Some bronchiolar epithelial cells displayed degeneration (Fig. 5b). No obvious histopathological change was found in lungs of WT-mice infected with pangolin-CoV-HKU4-P251T, or in control hDPP4-mice.

To verify inflammatory cells in pathological lesion areas, we tested the MAC2+ macrophages, CD3+ T lymphocytes, and CD19+ B lymphocytes using specific monoclonal antibodies[21]. Only prominent increases in MAC2+ macrophages were identified in the lungs of pangolin-CoV-HKU4-P251T-infected hDPP4-mice in comparison to the uninfected control hDPP4-mice (Fig. 5c). The MAC2+ macrophages were dispersed or aggregated in the alveolar cavities at 3 DPI, became diffusely infiltrated at 6 DPI, and were focally aggregated together in the thickened alveolar septa at 12 DPI. No obvious CD3+ T lymphocytes and CD19+ B lymphocytes were observed in the lung sections of the infected mice at any timepoints (Supplementary Fig. 5).

Cytokines have been considered to play an important role in the immune and immunopathology of viral infections[22], and prolonged proinflammatory response has been caused by MERS-CoV in a previous study[23]. To study the cytokine response in pangolin-CoV-HKU4-P251T infection, the mRNA expression levels of major antiviral

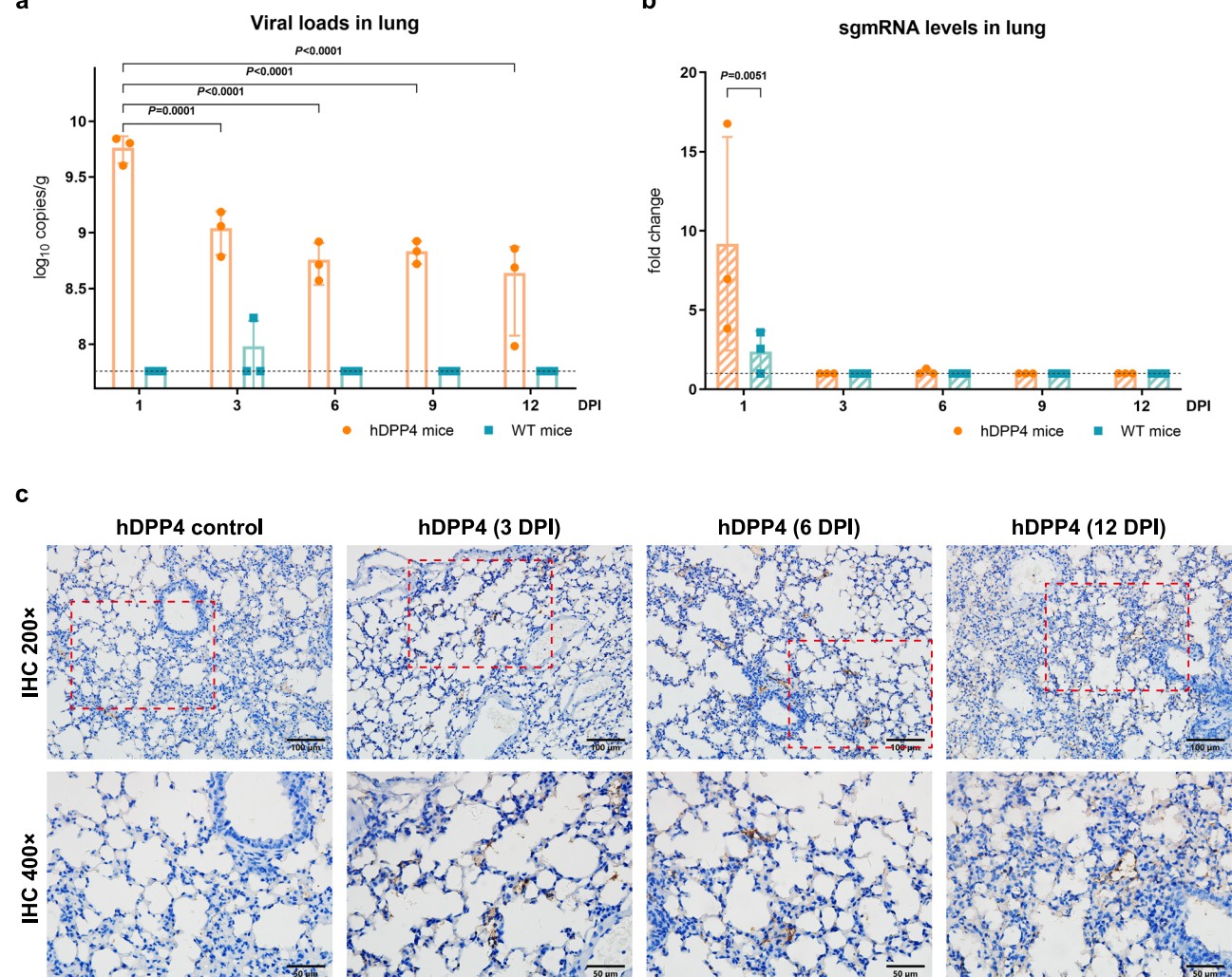

**Fig. 4 | Pangolin-CoV-HKU4-P251T infection in the lungs of hDPP4-transgenic mice. a** The copy number of viral RNA in lungs of pangolin-CoV-HKU4-P251T-infected hDPP4 (orange) and WT (blue) mice was determined using qRT-PCR. *n* = 3 biologically independent animals per genotype for each time point. Data are presented as mean ± SD with scatter plot at each time point. Dashed lines denote the detection limit. Whenever the sample measurement was below the detection limit, the result was assigned a value equal to the minimum detection limit to facilitate statistical analysis of the data. ANOVA was used for multiple comparisons. **b** Relative sgmRNA quantification in lungs of virus-infected hDPP4 (orange) and WT (blue) mice. *n* = 3 biologically independent animals per genotype

for each time point. Data are presented as mean ± SD with scatter plot at each time point. ANOVA was used for multiple comparisons. **c** The expression of viral nucleocapsid protein in lungs of infected hDPP4-mice. Representative IHC staining images of viral nucleocapsid protein expression in lungs of infected hDPP4-mice at 3, 6, and 12 DPI. The hDPP4-mice inoculated with cell medium are used as control. Images are representative of three experimental animals. Original magnification ×200 (upper panel). Red frames indicate regions shown in high magnification (lower panel, ×400). See also Supplementary Fig. 3 and 4a. Source data are provided as a Source Data file.

cytokines (IFN-β, IFN-γ, and Mx1), proinflammatory cytokines (IL-2, IL-12p40, and TNF-α), and chemokines (CXCL-1 and G-CSF) in lung samples of infected hDPP4-mice were measured using qRT-PCR assays (Supplementary Table 1) as described previously[16]. All the cytokines kept increasing in lung tissues from 3 to 12 DPI (Fig. 5d). The antiviral cytokines mainly including IFN-β, IFN-γ and Mx1 have been known to inhibit the entry of virus into cells, block viral replication, and indirectly stimulate innate and adaptive immune responses[24–26]. IL-2, IL-12p40, and TNF- α are key proinflammatory cytokines released by macrophages after coronavirus infections as the inflammatory responses, which can lead to lung cell damage[27,28]. The chemokines, CXCL-1 and G-CSF, which are expressed by macrophages, neutrophils, and epithelial cell, can attract other non-hematopoietic cells to the viral infection sites, playing an important role in regulating immune and inflammatory responses[29]. The persistent increases in main cytokines observed in lung tissues of the hDPP4-mice imply the prolonged persistence of viral RNA, which deserves further investigation.

## Discussion

We successfully isolated an unclassified species of *Merbecovirus*, pangolin-CoV-HKU4-P251T from a trafficked Malayan pangolin, the genome of which was most closely related to batHKU4-related coronaviruses from the greater bamboo bat as well as lesser bamboo bats. Pangolin-CoV-HKU4-P251T shows purifying selection in its S protein with only non-synonymous substitutions during passage in Vero 81 cells. Given that the passage 15 exhibited higher growth rates, the A843V substitution in the S protein might be associated with efficient virus replication, which deserves further investigation. The present findings not only provide evidence of pangolins as possible natural reservoir of *Merbecovirus* besides SARS-CoV-2-related serbecoviruses[2–10], but also highlight the genetically close relationship between pangolin-derived and bat-derived merbecoviruses. In addition, a member of subgenus *Merbecovirus* have been detected in European hedgehogs from Germany and France[30,31], and in Amur hedgehogs in China[32]. Considering bats, hedgehogs and pangolins are

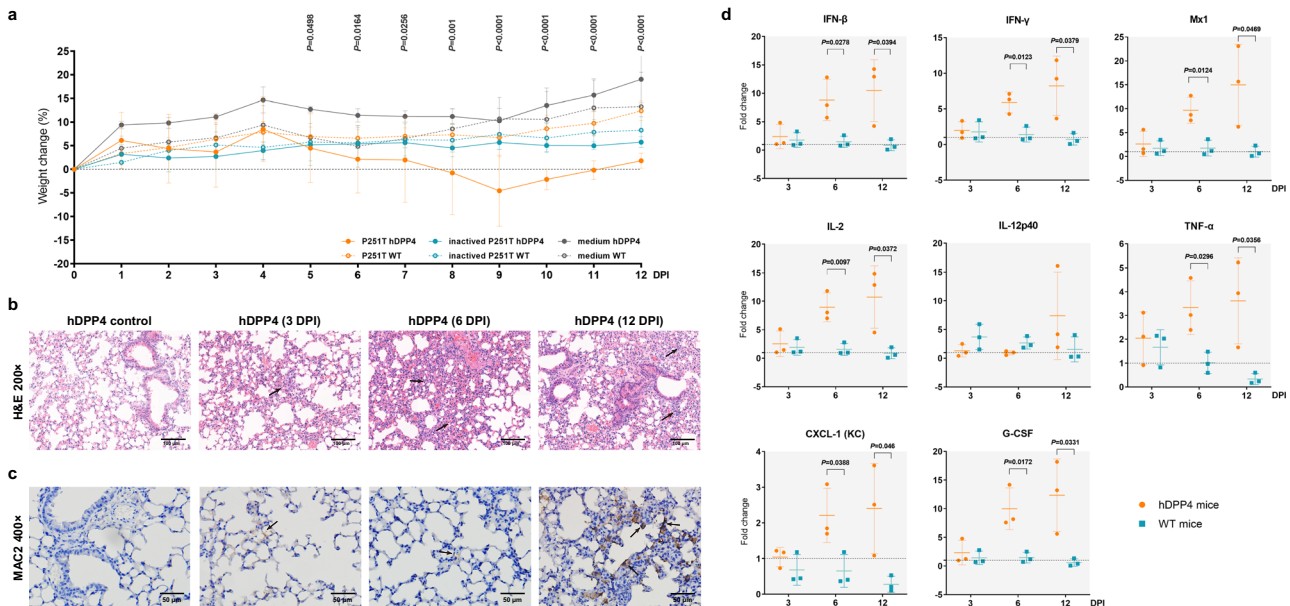

**Fig. 5 | Pathogenicity of virus-infected hDPP4-transgenic mice. a** Percentage of body weight changes of infected hDPP4-mice and WT-mice. $n = 17$ biologically independent animals per genotype. Data are reported as mean ± SD (shown as error bars). The solid dots and lines indicate the weight changes of hDPP4-mice. The hollow dots and dashed lines indicate the weight changes of WT-mice. Orange, live virus-infected mice; blue, heat-inactivated viruses-infected mice; dark gray, cell culture supernatant-infected mice. ANOVA was used for multiple comparisons. **b** Pathological features of virus-infected hDPP4-mice. H&E-stained lung sections at 3, 6, and 12 DPI are displayed. The hDPP4-mice inoculated with cell medium are used as control. Images are representative of three experimental animals. Original magnification ×200. See also Supplementary Fig. 4b. **c** Immunohistochemical

analysis of lung tissue stained with MAC2 antibody for macrophage detection. The hDPP4-mice inoculated with cell medium are used as control. Images are representative of three experimental animals. Original magnification ×400. See also Supplementary Fig. 4c. **d** Cytokine response in the lung of hDPP4 (orange) and WT (blue) mice following virus infection. mRNA levels were monitored by qPCR at 1, 3, 6, and 12 DPI. The relative expression of mRNA at 3, 6, and 12 DPI was measured by comparative $2^{-\Delta\Delta CT}$ method related to 1 DPI and the mean ± SD (shown as error bars) of fold change of three independent biological replicates for each gene at each time point are presented. Student's $T$-test (two-tailed) was used for comparisons of hDPP4- and WT-mice. Source data are provided as a Source Data file.

all insectivorous mammals, our findings further suggest that insectivorous mammals other than bats may contribute to the evolution and diversity of *Merbecovirus* including MERS-CoV.

The successful isolation of the specific coronavirus in subgenus *Merbecovirus* enable us to conduct in vitro and in vivo studies for understanding its receptor usage, infectivity and pathogenicity. Our findings reveal that pangolin-CoV-HKU4-P251T may be a potential emerging human pathogen through cross-species transmission. Firstly, the virus is capable to efficiently infect and replicate in multiple human cell lines, including Huh7, Calu-3 and Caco-2 cells, suggesting the susceptibility of humans to pangolin-CoV-HKU4-P251T infection. Secondly, pangolin-CoV-HKU4-P251T grows much better in hDPP4-expressing HeLa cells than in wild-type HeLa cells with 21.8-fold increase in viral copies, showing efficient utilization of hDPP4 receptor for infection. Thirdly, pangolin-CoV-HKU4-P251T is able to infect the hDDP4-transgenic mice, persist in the lungs, and cause interstitial pneumonia with prominent inflammatory changes. While the underline cross-species transmission mechanism and the role of host DPP4 remain to be elucidated, extensive surveillance on the pangolin-derived coronaviruses should be enhanced to reduce the public health threat to humans, and avoid possible risk for spillover events.

The growth of pangolin-CoV-HKU4-P251T is robust with sustainable viral copy rise in human Huh7, Calu-3, and Caco-2 cells besides primate Vero 81 cells (Fig. 2). Human hepato-cellular carcinoma Huh 7 and lung adenocarcinoma Calu-3 cells are known susceptible to a variety of human coronaviruses or related coronaviruses[33,34]. Caco-2 epithelial cells derived from human colorectal adenocarcinoma can be used for a broad range of respiratory viruses[35]. The efficient replication of pangolin-CoV-HKU4-P251T in the three human cell lines suggests the pangolin-borne *Merbecovirus* may infect humans, once it spills over in an adaptable circumstance. The suitable host receptors are critical for coronavirus

cross-species transmission from reservoir hosts to humans. Our findings reveal that pangolin-CoV-HKU4-P251T utilizes both pDPP4 and hDPP4 receptors proved by significantly increased viral replications in either pDPP4-expressing and hDPP4-expressing HeLa cells in comparison to wild-type HeLa cells (Fig. 3). These findings indicate that the virus not only utilizes the pangolin DPP4 receptor for infection in its natural hosts but also is able to use human DPP4 for possible cross-species transmission as an emerging human pathogen. Contrary to expectation, bDPP4-expressing HeLa cells do not show any effect on the replication of pangolin-CoV-HKU4-P251T, although it is closely related to bat-borne HKU4 coronaviruses. Moreover, pangolin-CoV-HKU4-P251T is unable to infect bat lung cells (Tb 1 Lu). While the sequence and binding analyses have indicated that the spike protein from bat HKU4 could recognize both hDPP4 and bDPP4 as its receptors[14,15], the experiments using live *Tylonycteris*-bat-CoV-HKU4 indicate that it can only use hDPP4 and dromedary camel-DPP4 (dcDPP4) as receptors, but not infect bDPP4-expressing cells or primary kidney and lung cells of *T. pachypus* bats, from which the virus was isolated[16]. Our findings together with those from the related study on *Tylonycteris*-bat-CoV-HKU4 isolate suggest that bat DPP4 may not be essential for *Tylonycteris*-Bat-CoV-HKU4 in its natural host, nor for its closely related merbecoviruses, such as pangolin-CoV-HKU4-P251T, from other animal hosts.

Coronavirus infections have already been shown to be age-dependent: The aged mice tend to be more susceptible to the viruses in subgenus *Sarbecovirus* such as SARS-CoV and SARS-CoV-2[36,37], while the experimental studies to evaluate the susceptibility of viruses in the subgenus *Merbecovirus* usually use immature mice[12,16]. Therefore, we select young mice for the in vivo experiments, and find that pangolin-CoV-HKU4-P251T shows its infectivity and pathogenicity in hDPP4-mice, characterized by the affected weight gain and interstitial pneumonia after intranasal inoculation. Either viral loads or antigens are sustained in

the lungs of infected hDPP4-mice during the observation period from 1 to 12 DPI, which are comparable to the results observed in the experimental study of Tylonycteris-bat-CoV-HKU4 using hDPP4-transgenic mice[16]. While the results of undetectable sgmRNA since 3 DPI in the lung samples indicate that there is no live virus thereafter. The positive results of qRT-PCR and IHC might be due to the residual viral nucleotides and proteins retained in the tissues. Moreover, we also found that virus-infected hDPP4-mice developed measurable neutralizing antibodies distinct from the WT-mice and any controls, which further demonstrated the infection of pangolin-CoV-HKU4-P251T in hDPP4-mice.

The progressive infiltration and aggregation of inflammatory cells, mainly macrophages, were observed in the infected lung tissues of hDPP4-transgenic mice. This kind of inflammatory response is something similar to that caused by SARS-CoV-2 infection in hACE2-transgenic mice, from which aggregated macrophages and only a few T lymphocytes or B lymphocytes were detected in the lung[21]. Why only macrophages predominately present in the inflammatory lung tissues of experimental mice after pangolin-CoV-HKU4-P251T or SARS-CoV-2 infection, is unclear and deserves further investigation. Notably, pangolin-CoV-HKU4-P251T infection in hDPP4-mice leads to prolonged increases in antiviral cytokines, proinflammatory cytokines, and chemokines in lung tissues as observed in *Tylonycteris*-bat-CoV-HKU4-infected hDPP4-mice[16], as well as in MERS-CoV infected cell cultures[23]. These findings suggest that the two closely related merbecoviruses from different animal hosts may share similar evolutionary trajectory, and have analogic pathogenesis as MERS-CoV.

One limitation of this study is that the infectivity and replication in primary human cells are unknown, although pangolin-CoV-HKU4-P251T grows well in multiple continuous human cell lines. Therefore, we cannot definitely conclude the cross-species transmission potential of the virus at this moment, because the cultures of primary and continuous cells contain different proteases, which may play a role in susceptibility to the virus. A further evaluation of virus growth in primary human airway or intestinal cells will provide more direct evidence of spill-over transmission possibility of the pangolin-associated virus. In addition, because of the limited observation time of the animal experiments, we were unable to obtain serological data of mice in the late stages of virus infection. Prolonged observation of animal experiments and real-time monitoring of live viruses will be required to further support the conclusion of productive infections.

Since the discovery of SARS-CoV-2-related coronaviruses in Malayan pangolins[2-4], several human-infecting viruses as well as a variety of other viruses have increasingly reported in smuggled or rescued pangolins[1,12,38-41]. Isolation of the coronaviruses is very important to do experiments for better understanding their cross-species transmissibility and potential for emergence as human pathogens. In the current study, we successfully isolate a virus in subgenus *Merbecovirus*, from a Malayan pangolin. Pangolin-CoV-HKU4-P251T infects hDPP4-transgenic mice leading to prominent inflammatory damages of lungs. Our findings provide further evidence similar to the results of a recent report using closely related isolate of MjHKU4r-CoV-1[12]. Both studies suggest that the pangolin-borne coronaviruses in subgenus *Merbecovirus* may be capable to spillover from pangolins to humans by using DPP4 receptors. While the underline mechanism of cross-species transmission remains to be fully investigated, our work further highlights the importance of pangolins in public health. People should keep away from the most trafficked wild animal in the world[42], to prevent potential pangolin-to-human transmission.

## Methods
### Ethics statement
All procedures involving virus and animals were conducted in biosafety level 3 laboratory (BSL-3) and approved by the Animals Experimental Committee, and Ethics Committee of Changchun Veterinary Research Institute (approval number: IACUC of AMMS-11-2022-011).

### Isolation of pangolin-CoV-HKU4-P251T
The original small intestine samples from Malayan pangolins, from which the whole genome sequence of pangolin-CoV-HKU4-P251T had been obtained (accession number OM009282)[1], were homogenized and propagated on African green monkey kidney epithelial cells Vero 81 (ATCC CCL-81). The small intestine samples were cut into small pieces, homogenized in serum-free DMEM medium followed by centrifugation at 800 g for 5 min, and the resulting supernatant was used for virus isolation. Mixed 1 mL of supernatant with 3 mL of fresh serum-free DMEM medium and seeded the mixture onto Vero 81 cell monolayers in a T75 flask. Following incubation at 37 °C for 1 h, the supernatant was removed and replaced with 12 mL of fresh DMEM with 2% FBS and 1% penicillin-streptomycin solution. Seven days post inoculation (DPI), the culture was frozen and thawed for 3 times, and the supernatant was collected for the stock. One mL supernatant was taken out from the stock, and inoculated into the naïve Vero 81 cells for the sub-cultivation using the same procedure described above. For each subsequent passage, viral loads were determined using the qRT-PCR method as described below. The viral genome of each passage was sequenced by next-generation sequencing to assess possible mutation of pangolin-CoV-HKU4-P251T through subcultures. The live virus was quantified using the 50% tissue culture infective dose (TCID$_{50}$) method. Cells were seeded in a 96-well plate and inoculated with serial dilutions of the virus. The cell plate was then incubated and examined for the virus-mediated cytopathic effect, and TCID$_{50}$/100 μL was calculated using the Reed-Muench method[43].

### Cell lines
The African green monkey kidney epithelial cells Vero 81, human cervical cancer cell line HeLa, and human hepatoma cell line Huh-7 were maintained in DMEM medium supplemented with 10% FBS. The human lung adenocarcinoma cell line Calu-3 and colorectal adenocarcinoma cell line Caco-2 were cultured in Minimum Essential Medium (MEM, Gibco, catalog 11095080) containing 10% and 20% FBS, respectively, supplemented with 1% Non-Essential Amino Acids Solution (Gibco, catalog 11140050) and 1 mM Sodium Pyruvate (Gibco, catalog 11360070). The bat lung cell line Tb 1 Lu were maintained in MEM medium supplemented with 10% FBS. The human bronchogenic carcinoma cell line ChaGo-K-1 was grown in Roswell Park Memorial Institute 1640 Medium (RPMI 1640, Gibco, catalog C11875500BT) containing 10% FBS. The human bronchial epithelial cell line BEAS-2B was incubated in Bronchial Epithelial Cell Growth Medium BulletKit (BEGM, LONZA, catalog CC-3170). All cells were grown at 37 °C in a humidified 5% CO$_2$ atmosphere. The hDPP4 and hACE2 expression in different cell lines was determined by western blotting assay using Anti-DPP4 antibody (Abcam, ab215711, 1:1000 dilution), Anti-ACE2 antibody (Abcam, ab108209, 1:1000 dilution), HRP-labeled Goat Anti-Rabbit IgG(H + L) (Beyotime, A0208, 1:1000 dilution).

### Experimental animals
hDPP4-transgenic mice and wild-type C57BL/6 mice were used in this study. Five weeks old hDPP4 female mice (Cat. NO. NM-HU-190042) were purchased from Shanghai Model Organisms Center, Inc. (https://www.modelorg.com/). The DPP4 targeting construct was designed to insert a DPP4 gene and a FLAG and a SV40 polyA signal into exon3 of the targeted gene. This construct was inserted into the targeted gene via CRISPR/Cas9 system in C57BL/6 mouse background. The donor vector with sgRNA and Cas9 mRNA were microinjected into C57BL/6 fertilized eggs. F0 generation mice positive for homologous recombination were identified by long PCR. The PCR products were further confirmed by sequencing. F0 mice were crossed with C57BL/6 mice to obtain *Dpp4*$^{+/DPP4}$ heterozygous mice. Five weeks old wild-type female mice were obtained from Beijing Weitong Lihua Biotechnology Co., Ltd. All animals were housed at the conventional clean animal facility of the animal biosafety level-3 (ABSL-3). The housing environment

included a 12-h light / dark cycle with constant room temperature (22–24 °C), humidity (45–65%), and free access to water and diet. All animal experiments were approved by the Institutional Animal Care and Use Committee of the Institute of Military Veterinary Medicine, Academy of Military Medical Sciences (IACUC of AMMS-11-2022-011).

## Infections in different cell lines

To compare the cell tropism and virus replication competence of pangolin-CoV-HKU4-P251T in different cell lines, including Huh7, Caco-2, Calu-3, BEAS-2B, ChaGo-K-1 and Tb 1 Lu, we analyzed the growth kinetics of the viruses. Viral growth curves were performed by inoculating with viruses at MOI of 0.01 in triplicate wells of a 12-well plate. After 1 h of incubation at 37 °C, the cells were gently washed with PBS twice and continue cultured in fresh medium. The supernatants were collected at 0, 12, 24, 48, and 72 HPI for RNA extraction and viral load quantification.

## Infection assay in DPP4-expressing cells

Complementary DNA (cDNA) containing coding sequence of hDPP4, pDPP4, bDPP4, hACE2, pACE2, and bACE2 were synthesized by Sango Biotech (Shanghai, China), and cloning into the KpnI/XhoI (NEB, America) sites of pcDNA3.1-HA plasmid. Transfection of plasmids into HeLa were conducted using the Lipofectamine™ 3000 according to manu-facture's instruction. HeLa cells were lysed in RIPA buffer (Beyotime, China) with protease inhibitor cocktail on ice for 30 min. After the supernatant was harvested and total protein concentration was deter-mined by a BCA protein assay kit (Beyotime, China), protein lysates were separated by SDS-PAGE and transferred to polyvinylidene difluoride (PVDF) membranes via semidry transfer. The membranes were blocked with 5% BSA for 1 h at room temperature before incubating with specific HA antibodies (Biolegend, MMS-101P, 1:500 dilution), followed by HRP-labeled Goat Anti-Mouse IgG (H + L) (Beyotime, A0216, 1:1000 dilution).

Cells grown in 12-well plates were infected with the virus at MOI of 0.01. After virus adsorption, cells were washed with PBS twice and further incubated in fresh medium at 37 °C. Meanwhile, the HeLa cells were used as a negative control and each experiment was performed in three biological replicates. The supernatants were harvested at 48 HPI and viral load were measured by qRT-PCR.

## Animals infection experiments

The experimental groups of hDPP4-transgenic mice and wild-type C57BL/6 mice were inoculated intranasally by gently adding 50 μL droplets of virus stock with a viral titer of $10^4$ TCID$_{50}$. The control groups received an equivalent fluid volume of cell culture supernatant or heat-inactivated virus. After inoculation, and throughout the entire course of the study, mice were weighed and observed daily for clinical signs of disease. For the experimental group of pangolin-CoV-HKU4-P251T, three mice were euthanized on days 1, 3, 6, 9 and 12 post-infection. After euthanasia, the mice were necropsied and tissues samples were collected, including nasal turbinate, trachea, lung, heart, brain, kidney, spleen, liver, and small intestine. The hDPP4 expression was detected using western blotting assay in all the collected tissues of both hDPP4-transgenic and wild-type mice. Total RNA was extracted for viral load analysis by qRT-PCR. For histopathology, the tissue of 3, 6, 12 DPI was fixed in 4% paraformaldehyde and paraffin-embedded.

## RNA extraction and quantitative real-time PCR

To quantify viral load, total RNA was extracted from the collected samples using QIAamp Viral RNA Mini Kit (Qiagen, catalog 52906) according to the manufacturer's instructions. The purified RNA was then used for quantitative real-time RT-PCR (qRT-PCR) amplification targeting the ORF1a regions of the pangolin-CoV-HKU4-P251T genome using the One Step TB Green PrimeScript RT-PCR Kit (TaKaRa, RR066A). Furthermore, the presence of live viruses in the lungs of mice was measured by qRT-PCR of E gene subgenomic mRNA. Mouse β-Actin was applied as an internal control gene. For relative

quantification, $2^{-\triangle\triangle Ct}$ was calculated and used as an indication of the relative expression levels[44]. The related primer sequences were dis-played in Supplementary Table 1.

Primer sequences are shown in Supplementary Table 1. All qRT-PCR was performed on a CFX96 Real-Time PCR Detection System (Bio-Rad) with three technical replicates. The copy number of viral RNA was calculated using a standard curve.

## Negative stain Electron microscopy

For morphological and ultrastructural observation of virus, the virions were viewed under transmission electron microscopy (TEM) using the conventional negative staining procedure. Regarding the preparation of electron microscope samples, scraped the host cells infected with virus using a cell scraper and collected them in a tube, then centrifuged at 100 g for 5 min. Supernatants containing virions were harvested and fixed in 2.5% glutaraldehyde at 4 °C overnight, then putting a droplet of the treated solution on a copper grid coated with a thin carbon film for 5 min and negatively stained with 2% phosphotungstic acid for 3 min. Sample observations were conducted and photographs were taken using a JEOL JEM-1200 transmission electron microscope operated at 80 kV.

## Fluorescence in situ hybridization (FISH)

Custom Stellaris FISH Probes were designed by utilizing the Stellaris RNA FISH probe Designer (Biosearch Technologies Inc., Petaluma, CA) available online at www.biosearchtech.com/stellarisdesigner to detect the ORF1ab gene of pangolin-CoV-HKU4-P251T, and were cou-pled to Quasar 570 (Supplementary Table 2). A set of Stellaris RNA FISH Probes is comprised of up to 48 singly labeled oligonucleotides designed to selectively bind to targeted transcripts. Cells cultured on sterile microscope cover glass in 12-well plates were infected by 0.01 MOI virus for 48 h. Afterward, cells were fixed with 4% paraformalde-hyde for 15 min at room temperature and were washed 3 times with phosphate-buffered saline (PBS). To permeabilize the cells, the cells were placed in 70% ethanol and incubated for at least 1 h at 4 °C. The hybridization and washing were performed according to the manual of Biosearch Technologies available online at www.biosearchtech.com/stellarisprotocols. All treated samples including control cells were visualized by fluorescent microscope (Olympus).

## Indirect immunofluorescence assay (IFA)

Indirect immunofluorescence assay was used to verify the intracellular expression of receptor protein and the localization of the viral protein. Purified anti-HA.11 Epitope Tag Antibody (BioLegend, Catalog MMS-101P) and Rabbit anti-Pangolin-CoV-HKU4-P251T Nucleoprotein Pab custom-made by Sino Biological Inc. were used as primary antibodies. The secondary antibody included goat anti-mouse IgG Alexa Fluor 488 (Abcam, ab150117) and goat anti-rabbit IgG Alexa Fluor 594 (Abcam, ab150080). Cells in 12-well plates were infected with the viruses at an MOI of 0.01 for 48 h. After three washes with PBS, the cells were fixed with 4% paraformaldehyde for 15 min at room temperature and treated for 1 h with 70% ethanol at 4 °C to permeabilize the cell membranes. After rinsing with PBS three times, the cells were incubated in blocking buffer (5% bovine serum albumin in PBS) for 1 h at room temperature to block non-specific binding. After removal of the blocking buffer, primary antibodies (1:100 dilution) were added and incubated at 37 °C for 1 h, followed by washing and incubation with the secondary anti-body (1:200 dilution) at 37 °C for 45 min. Images were observed and captured by fluorescent microscope (Olympus).

## Histopathological analysis

To examine the histopathology of viruses in tissues from infected mice, the mice lung of 3, 6, 12 DPI were subjected to hematoxylin-eosin (H&E) and immunohistochemical (IHC) staining as previously described[21]. In brief, the harvested tissues were fixed with 4%

paraformaldehyde, dehydrated, embedded in paraffin, cut into 4 μm sections, and stained with H&E. Pathological examination by IHC staining were conducted with the pangolin-CoV-HKU4-P251T Nucleoprotein Pab (Sino Biological Inc., customized, 1:100 dilution), MAC2 antibody (Cedarlane Laboratories, CL8942AP, 1:1000 dilution), CD3 antibody (Sino Biological Inc., 108567-T08, 1:500 dilution) and CD19 antibody (Cell Signaling Technology, 3574, 1:50 dilution). H&E and IHC staining was performed in a blind manner and pathologists who performed the staining and observation did not know the original hypothesis.

### Cytokines analysis
In order to examine if infection with pangolin-CoV-HKU4-P251T may induce the immune response, we detected the cytokine expression in the lung of virus-infected mice at 1, 3, 6, and 12 DPI. The purified RNA of lung was used for qRT-PCR amplification targeting the gene of interferons (IFN-β, IFN-γ), Mx1, interleukins (IL-2, IL-12p40), TNF-α, CXCL-1 (KS), and G-CSF using the One Step TB Green PrimeScript RT-PCR Kit (TaKaRa, RR066A). Relative gene expression was determined by normalizing the gene expression of each target gene to β-actin. The related primer sequences were displayed in Supplementary Table 1. The relative expression of mRNA at 3, 6, and 12 DPI was measured by comparative $2^{-\Delta\Delta CT}$ method related to 1 DPI[44].

### Neutralization assay
Both hDPP4- and WT-mice serum were collected at each time point after virus infection to measure serum neutralizing antibody titers. Huh7 cells were seeded in 96-well plates and incubated overnight at 37 °C under 5% $CO_2$ to allow formation of cell monolayers. All individual serum were heat-inactivated for 30 min at 56 °C, serially diluted (2-fold dilutions starting at 1:10), and were assayed against 100 $TCID_{50}$ of virus in 96-well plates (1:1 mixtures). The control mice serum and cell culture medium served as negative control and the pure virus as positive control. The plates were incubated at 37 °C for 5 days, and cytopathic effects were observed. Serum neutralizing antibody titers were expressed as 50% neutralizing antibody titers (NT50).

### Sequence, phylogenetic and recombination analysis
Quality control and sequencing of total RNA were completed by Novogene Bioinformatics Technology Co., Ltd (Beijing, China). Libraries were constructed by Novogene and sequenced with the Illumina HiseqX-ten PE150 platform. The sequenced raw data were filtered and the adapter sequence and low-quality data were removed by the AfterQC (v0.9.7)[45], resulting in the clean data used for subsequent analysis. Then, these reads were aligned to the host reference genome, and duplicates were removed to obtain valid data without host contamination. Next, the remaining valid data were assembled into a de novo transcriptome using Trinity software (v2.8.5) set to default parameters[46]. All the sequences collected from GenBank were first aligned using the program MAFFT (v7.505)[47]. All ambiguously aligned regions were then removed using the TrimAl program[48]. For each sequence alignment, phylogenetic trees were then inferred using the maximum likelihood approach (ML) implemented in IQ-Tree (v2.2.0.3)[49]. The ML trees were visualized using ggtree package (v3.6.2)[50]. To investigate potential recombination events, we used SimPlot (v3.5.1)[51] to conduct a window sliding analysis to determine the changing patterns of sequence similarity between the query (OM009282) and the reference sequences.

### Statistical analysis
All statistical analyses were performed using GraphPad Prism 7.0 software. The data are expressed as mean ± SD and were analyzed for statistical significance by the two-way ANOVA with correction to compare multiple groups of data. Two-sided Student's $T$-test was used to compare two independent groups. $P$ values that were ≤0.05 were considered statistically significant (0.01 <*$P$ ≤ 0.05; 0.001 <**$P$ ≤ 0.01; 0.0001 <***$P$ ≤ 0.001, ****$P$ ≤ 0.0001).

### Reporting summary
Further information on research design is available in the Nature Portfolio Reporting Summary linked to this article.

## Data availability
The high-throughput sequencing data used in this study are available in the Sequence Read Archive database (https://www.ncbi.nlm.nih.gov/sra) under accession code SRR25655199-SRR25655213 (Supplementary Table 3). Source data are provided as a Source Data file. All other relevant data are available within the article and Source Data file. Source data are provided with this paper.

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

## Acknowledgements

This research was supported by the National Natural Science Foundation of China (NSFC grant no. 81621005, W.-C.C.) and the State Key Research Development Program of China (grant no. 2019YFC1200501, J.-F.J.).

## Author contributions

W.-C.C., Y.-W.G., and J.-F.J. designed and supervised the research. L.-Y.X., Z.-F.W., Y.-G.L., D.-Y.Z., F.-X.L., J.Z., M.-Z.Z., W.-Y.G., L.-F.L., W.-H.W., N.J. and T.-C. W. carried out experiments. X.-M.C. and T.-C.Q. collected and processed samples. L.-Y.X., Z.-F.W. and R.-Z.Y. performed data analysis, the genomic and evolutionary analysis. W.-C.C., J.-F.J., Y.-W.G. L.-Y.X., Z.-F.W., coordinated data analysis and wrote the paper. All authors reviewed and edited the manuscript.

## Competing interests

The authors declare no competing interests.
