## [Peer Review File · Nature Communications]

REVIEWER COMMENTS

Reviewer #1 (Remarks to the Author):

The manuscript by Xia et al., "A pangolin-borne HKU4-related coronavirus infects human-DPP4-transgenic mice" describes the in vitro cultivation and in vivo pathogenesis of a pangolin Merbecovirus HKU4 in hDPP4 transgenic mice. The topic is important and timely and the authors convincingly show virus growth in vitro, although several research gaps remain that limit the impact of these studies. However, additional research is needed to support and expand upon the preliminary findings. Moreover, a variety of issues exist within the manuscript that must be addressed prior to acceptance of this work.

Specific Comments:

- 1) The introduction and discussion should include references and text noting the recent publication of a HKU4-like PgCoV (MjHKU4r-CoV) in Cell (Chen J et al., Cell. 2023 Feb 16;186(4):850-863.e16.). How close are these two viruses in terms of overall and spike sequence and amino acid identity?
- 2) Figure 1, Genome Sequence is provided for passage 1 material in Vero Cells, not the adapted virus that replicated most efficiently after passage 13. The RNAseq data through passage 15 (not just the two adaptive changes) should be provided as this provides important information on virus evolution in culture.
- 3) Line 14. Is p1 or in vitro evolved Pangolin-CoV-HKU4-P251T being used in these experiments? How is the in vitro passage virus stock designated differently from the original virus inoculum isolated from pangolins? Is the in vitro passaged stocks used throughout?
- 4) Figure 1, a ~75 fold increase in genome copies could equate to a minor increase in infectious virus released from these cultures. It is critical to perform an actual growth curve of live virus comparing the p1 and in vitro adapted viruses.
- 5) The MOI used in these growth curves is unclear, please provide this information. Also, time 0 and 72 hr live virus titrations need to be performed in at least one of the cell lines, I would recommend Huh7 as they had the greatest increase in genome copy number. Are ACE2 and DPP4 both expressed in these cell lines?
- 6) Figure 3a/3b. Some Merbecoviruses can use ACE2 receptors for docking and entry. It is important to determine whether or not P251T can use ACE2 receptors for entry using similar assays. Also, what is being merged in figure 3? DAPI and viral antigen? Shouldn't this figure be showing co-localization of DPP4 receptor expression and viral antigen? Please redo.
- 7) Mouse studies are being performed with a viral load of 10e9. What is the actual virus titer (either by TCID50, Focus forming assays or PFU)? Virus growth is not overwhelming and the primary argument is that genome equivalents decrease over time (Fig 4). What about real time detecting evidence of viral subgenomic mRNA in the lungs of these animals at later timepoints? The viral antigen data is interesting, yet long-term persistence of viral antigen in mice is unexpected, as other Merbecoviruses clear fairly quickly. Standing alone, the antigenic data is not convincing as it could represent background staining.

Have the authors sequenced the material at later timepoints, perhaps showing that the virus is adapting to the host?

8) The source of the DPP4 transgenic mice is unclear in the methods of the manuscript. How is expression driven? Has this mouse resource been reported previously and if not, has hDPP4 expression been evaluated in various tissues of these animals?

9) Lines 193-205. PgCoV P251T mice did not lose weight after infection (Fig 5). The DPP4 transgenic animals didn't gain weight like controls, which is a different phenotype. This section needs to be written to accurately reflect the phenotype. Weight loss statements are simply incorrect and inappropriate. It is strange that only macs are increasing. No T cell responses or antibody responses in these animals by day 12?

10) Discussion.

a. Only no-synonymous mutations over 15 passages? Include the RNA seq data for several different passages as reviewers need to be able to examine these data directly as it seems unlikely and needs to be supported with raw read data.

b. The authors argue that the P251 virus may not use DPP4 in its natural host. Given the recent published literature of ACE2 using Merbecoviruses, they should test whether its using ACE2 in its natural host

c. Discussion throughout. The virus doesn't cause any body weight loss. It prevents body weight gain. A very different phenotype and it needs to be discussed properly in the discussion. As currently written, the statement is inaccurate.

d. Line 287 "may be" is a more appropriate verb usage here, as the authors haven't demonstrated virus growth in primary human airway or intestinal cells. A pre-requirement is showing such data, alternatively the limitation of the study should be directly acknowledged. Primary and continuous cells in culture are different, as are proteases, which may play a role in cross species transmission and replication.

e. Lines 308+ significant body weight loss needs to be changed to prevent significant increases in body weight gain after infection (throughout discussion).

f. Line 326 or so. A recent publication in Cell describes a Merbecovirus that uses hDPP4 receptors and grows in mice. It needs to be acknowledged and discussed in the discussion. What are the similarities and differences in the two studies.

Reviewer #2 (Remarks to the Author):

In this manuscript, Xia et al. isolated an HKU4-related coronavirus – pangolin-COV-HKU5-P251T – from a Malayan pangolin. The authors determined that this newly identified virus shares 94.3% nucleotide identity with a coronavirus isolated from the greater bamboo bat (*Tylonycteris robustula*). Furthermore, the virus was characterized in vitro and in vivo. The authors demonstrated the ability of the virus to human DPP4 to successfully infect human cell lines. In addition, a primary characterization in DPP4-expressing mice was provided.

Overall, the study by Xia et al., is of importance, as it demonstrates the existence of a zoonotic with the potential to jump into the human population and to cause disease.

A few minor comments that will help to improve the manuscript.

The manuscript is not as well written as articles traditionally published in this publication. It is at times difficult to follow, which detracts from the science. The authors need to improve the language with regard to grammar and word usage, as well as general flow.

Several passages that are currently in the Results section should go into the Discussion section (e.g. line 134-138, line 109-110)

What is the rationale for the claim in Line 109- 110 that ‘the A843V

substitution in spike protein might be associated with viral activity’? - This should also be in the Discussion, including their analysis in reaching the conclusion, and other possible explanations, if any. [For example, could this be an artifact?]

Line 123

Here, the authors conclude that 'the signal intensity at 48 HPI in each

cell line was in accordance with the viral load result detected by qRT-PCR’. – To compare staining in cells to a PCR signal, the authors need to quantify the signal strength of the intensity of the FISH signal in the individual samples.

What was the rationale for using immunologically immature mice? The authors should address this decision, as the age of the mice could have influenced the results shown. Specifically, Coronavirus infection has already been shown to be age-dependent. The authors should consider an additional experiment in aged mice; this model might develop pathology. At least, this possibility should be addressed in the Discussion.

Reviewer #4 (Remarks to the Author):

The authors identified an HKU4 related coronavirus from a Malayan pangolin. They minced intestinal tissue from the pangolin and inoculated Vero and other cell lines. They detect viral genome signal and FISH staining for orf1ab in some cell types that they describe as efficient replication. However, they do not describe any cytopathic effect and the increase in genome copies is less than 100-fold. Infection of hDPP4 transgenic mice is not convincing. There is no evidence of an increase in genome replication in the lungs of mice; there is a slower weight gain but no weight loss and the histologic changes are not convincing evidence of viral replication. The authors should consider inoculating a control group of animals with heat or UV inactivated virus.

Minor comments:

1. Line 52: change the word denominated to named
2. Line 54: in silico

Responses to Reviewer #1:

1) The introduction and discussion should include references and text noting the recent publication of a HKU4-like PgCoV (MjHKU4r-CoV) in Cell (Chen J et al., Cell. 2023 Feb 16;186(4):850-863.e16.). How close are these two viruses in terms of overall and spike sequence and amino acid identity?

Response: We appreciate the reviewer's comments, and have included the reference in Introduction and Discussion of the revised manuscript as following: "Another paper also reported a genetically close MjHKU4r-CoV from the pangolin anal swab samples¹²" (Page 3, Line 53-54) "Our findings provide further evidence similar to the results of a recent report using closely related isolate of MjHKU4r-CoV-1¹². Both studies suggest that the pangolin-borne coronaviruses in subgenus *Merbecovirus* may be capable to spillover from pangolins to humans by using DPP4 receptors." (Page 20, Line 417-421)

In response to the reviewer's inquiry, we have compared the sequence identities of the closely related coronaviruses in Results of the revised manuscript as following: "Our pangolin-CoV-HKU4-P251T genome (GenBank accession No. OM009282.1) was 30,344 bp in length, and shared 99.1% nucleotide (nt) identity with the genome sequence of MjHKU4r-CoV-1 from a Malayan pangolin (GenBank accession No. OQ786861.1)¹², and 94.3% nt identity with a coronavirus from the greater bamboo bat (*Tylonycteris robustula*) at Yunnan Province, China (GenBank accession No. ON745165.1), which was released in GenBank on Aug. 15, 2022 (Supplementary Fig. 1a). The amino acid (aa) identity in RNA-dependent RNA polymerase (RdRp) was 99.8% with MjHKU4r-CoV-1 and 98.7% with the coronavirus from a greater bamboo bat, which was higher than the previously reported 96.7% between pangolin-CoV-HKU4-P251T and *Tylonycteris*-bat-CoV-HKU4 from lesser bamboo bats at Hong Kong, China¹. The spike protein of pangolin-CoV-HKU4-P251T showed 99.3% aa identity with MjHKU4r-CoV-1, and 91.8% with the coronavirus from the greater bamboo bat. The current findings indicate that pangolin-CoV-HKU4-P251T together with MjHKU4r-CoV-1 are genetically closest to the coronavirus from greater rather than lesser bamboo bats." (Page 5-6, Line 90-106)

Accordingly, we have reconstructed the phylogenetic trees by including

MjHKU4r-CoV-1 sequence (Supplementary Fig. 1a).

2) Figure 1, Genome Sequence is provided for passage 1 material in Vero Cells, not the adapted virus that replicated most efficiently after passage 13. The RNAseq data through passage 15 (not just the two adaptive changes) should be provided as this provides important information on virus evolution in culture.

Response: We appreciate the reviewer's suggestion, and have provided the sequence data through 15 passages by adding the Fig. 1d and described the substitutions during subcultures in the revised manuscript as following: "We conducted viral sequencing of the 15 passages to assess possible mutation of pangolin-CoV-HKU4-P251T through subcultures (Supplementary Table 3), and found that it had a persistent synonymous substitution at position 2702 (AGC:AGT) of ORF1ab and a consistent non-synonymous substitution of A2P in NS3d protein from the fourth to 15th passage. A non-synonymous substitution leading to A843V in spike protein occurred since the fifth passage (Fig. 1d). These findings indicate the virus was rather stable during serial passages *in vitro*." (Page 6, Line 116-124)

3) Line 74. Is p1 or in vitro evolved Pangolin-CoV-HKU4-P251T being used in these experiments? How is the in vitro passage virus stock designated differently from the original virus inoculum isolated from pangolins? Is the in vitro passaged stocks used throughout?

Response: We appreciate the reviewer's inquiry, and have clarified the virus stock used in the subsequent experiments in Results of the revised manuscript as following: "Considering that pangolin-CoV-HKU4-P251T maintained stability during serial passage with only two non-synonymous mutations in the genome, the 15th passage viruses were used in the subsequent experiments due to the availability of a high-titer stock." (Page 6, Line 125-128)

In addition, we have provided a detail description about the *in vitro* cultivation passage in Methods of the revised manuscript as following: "Seven days post inoculation (DPI), the culture was frozen and thawed for 3 times, and the supernatant was collected for the stock. One mL supernatant was taken out from the stock, and inoculated into the naïve Vero 81 cells for the sub-cultivation using the same procedure described above. For each subsequent passage, viral loads were determined using the qRT-PCR method as described below. The viral

genome of each passage was sequenced by next-generation sequencing to assess possible mutation of pangolin-CoV-HKU4-P251T through subcultures.” (Page 30-31, Line 642-649)

4) Figure 1, a ~75 fold increase in genome copies could equate to a minor increase in infectious virus released from these cultures. It is critical to perform an actual growth curve of live virus comparing the p1 and *in vitro* adapted viruses.

Response: We are grateful for the valuable comment, and have conducted an additional experiment to evaluate the kinetics of virus replication of original isolate and adapted virus. We have added the results in the revised manuscript as following: “Furthermore, we compared the kinetics of virus replication between the original isolate (passage 1) and *in vitro* adapted strain (passage 15) that used for the experiments in this study. The adapted viruses grew more efficiently in the initial stage reaching peak at 48 HPI, and slowed down thereafter. While the titer of the originally isolated virus was gradually increased and up to the comparable level with the adapted virus at 144 HPI (Fig. 2f).” (Page 8, Line 159-164)

5) The MOI used in these growth curves is unclear, please provide this information. Also, time 0 and 72 hr live virus titrations need to be performed in at least one of the cell lines, I would recommend Huh7 as they had the greatest increase in genome copy number. Are ACE2 and DPP4 both expressed in these cell lines?

Response: We appreciate the reviewer’s reminding, and have clarified the viral MOI used in growth curves in the Methods of the revised manuscript as following: “Viral growth curves were performed by inoculating with viruses at MOI of 0.01 in triplicate wells of a 12-well plate.” (Page 33, Line 696-697) “Cells grown in 12-well plates were infected with the virus at MOI of 0.01.” (Page 34, Line 715)

As recommended by the reviewer, we have performed an experiment to quantify the live viral titrations using Huh7 cell line, and added the result to the revised manuscript as following: “Notably, obvious cytopathic effect (CPE) was visualized in the pangolin-CoV-HKU4-P251T-infected Huh7 cells (Fig. 2d), while CPE was subtle in either infected Vero 81 or Caco-2 and Calu-3 cells. Therefore, we used Huh7 cell line to quantify the viral growth and activity using the method of 50% tissue culture infective dose (TCID₅₀). The live virus titer was continuously increased from 24 to 72 HPI (Fig. 2e), indicating the persistent growth in the culture.” (Page 8, Line 154-159)

In response to the inquire about the receptors of the cell lines used in this study, we have done related experiment and added the results in the revised manuscript as following: “To investigate if the differences in viral replication were possibly related to the expression of hDPP4 and hACE2 receptors among the cell lines, we conducted the western blotting assay. The hDPP4 expression was observed in all four cell lines susceptible to pangolin-CoV-HKU4-P251T. While hACE2 was expressed in Vero 81, Caco-2 and Calu-3 cells, but not in Huh7 cell line, on which the virus most efficiently replicated (Fig. 2c). These findings suggest that the potential spill-over transmission of the virus might depend on hDPP4 rather than hACE2.” (Page 7-8, Line 146-153)

6) Figure 3a/3b. Some Merbecoviruses can use ACE2 receptors for docking and entry. It is important to determine whether or not P251T can use ACE2 receptors for entry using similar assays. Also, what is being merged in figure 3? DAPI and viral antigen? Shouldn't this figure be showing co-localization of DPP4 receptor expression and viral antigen? Please redo.

Response: We appreciate the reviewer's valuable suggestions, and have performed supplementary experiments. The results have been provided to the revised manuscript as following: “To further investigate whether pangolin-CoV-HKU4-P251T can utilize DPP4 and ACE2 as receptors for infection, we conducted the experiments by observing the virus infectivity in wild type HeLa (HeLa-WT) cells as well as human DPP4-expressing HeLa (HeLa-hDPP4), *Manis javanica* pangolin DPP4-expressing HeLa (HeLa-pDPP4) and *T. pachypus* bat DPP4-expressing HeLa (HeLa-bDPP4), human ACE2-expressing HeLa (HeLa-hACE2), pangolin ACE2-expressing HeLa (HeLa-pACE2) and bat ACE2-expressing HeLa (HeLa-bACE2) cells (Fig. 3a and c). Pangolin-CoV-HKU4-P251T grew well and replicated efficiently in hDPP4-HeLa and pDPP4-HeLa cells, with 21.8 and 64.5-fold increase in viral copies, respectively, in comparison to viral copies in HeLa-WT cells at 48 HPI. The fold increase in viral copies was 5.0 in HeLa-bDPP4 cells, which was significantly lower than those in HeLa-hDPP4 and HeLa-pDPP4 cells (Fig. 3b). In contrast, ACE2-expressing cells did not show any increase in viral load compared to HeLa-WT cells, suggesting that ACE2 may not be an essential receptor for pangolin-CoV-HKU4-P251T (Fig. 3d).” (Page 9, Line 178-191)

In addition, we have also given a rational explanation for the addition experiment concerning ACE2 in the introduction section in the revised manuscript as following: “Otherwise, a recent study found that MERS-CoV-related viruses from bats can use angiotensin-converting enzyme 2 (ACE2) as an entry receptor¹⁸.”

(Page 4, Line 64-66)

In response to the reviewer’s concern about viral antigen, we observed the co-localization images of receptor expression and viral antigen, and revised both Figure 3e-f and the description of the results as following: “We then examined the pangolin-CoV-HKU4-P251T-infected cells using indirect immunofluorescence assay (IFA), and observed obvious viral fluorescence signals in HeLa-hDPP4 and HeLa-pDPP4 cells, and only weaker fluorescence signals in HeLa-bDPP4 cells (Fig. 3e). No viral signals were observed in any kinds of ACE2- expressing cells (Fig. 3f). These findings indicate that pangolin-CoV-HKU4-P251T can efficiently utilize both hDPP4 and pDPP4 receptors rather than bDPP4 receptor to infect cells, further implying its adaptation to pangolins possibly as the reservoir hosts and potential infectivity to humans by using DPP4 rather than ACE2 receptors.” (Page 9-10, Line 192-201) Relevant experimental methods have also been added to the Methods as following: “Indirect immunofluorescence assay was used to verify the intracellular expression of receptor protein and the localization of the viral protein. Purified anti-HA.11 Epitope Tag Antibody (BioLegend, Catalog MMS-101P) and Rabbit anti-Pangolin-CoV-HKU4-P251T Nucleoprotein Pab custom-made by Sino Biological Inc. were used as primary antibodies. The secondary antibody included goat anti-mouse IgG Alexa Fluor 488 (Abcam, ab150117) and goat anti-rabbit IgG Alexa Fluor 594 (Abcam, ab150080). Cells in 12-well plates were infected with the viruses at an MOI of 0.01 for 48 hours. After three washes with PBS, the cells were fixed with 4% paraformaldehyde for 15 min at room temperature and treated for 1 hour with 70% ethanol at 4°C to permeabilize the cell membranes. After rinsing with PBS three times, the cells were incubated in blocking buffer (5% bovine serum albumin in PBS) for 1 hour at room temperature to block non-specific binding. After removal of the blocking buffer, primary antibodies (1:100 dilution) were added and incubated at 37°C for 1 hour, followed by washing and incubation with the secondary antibody (1:200 dilution) at 37°C for 45 min. Images were observed and captured by fluorescent microscope (Olympus).” (Page 37, Line 781-796)

7) Mouse studies are being performed with a viral load of 10e9. What is the actual virus titer (either by TCID50, Focus forming assays or PFU)? Virus growth is not overwhelming and the primary argument is that genome equivalents decrease over time (Fig 4). What about real time detecting evidence of viral subgenomic mRNA in the lungs of these animals at later timepoints? The viral antigen data is interesting, yet long-term persistence of viral antigen in mice is unexpected, as other Merbecoviruses clear fairly quickly. Standing alone, the antigenic data is not convincing as it could represent

background staining. Have the authors sequenced the material at later timepoints, perhaps showing that the virus is adapting to the host?

Response: In response to the reviewer's concern about the viral load, we have used the retested viral titer using TCID₅₀ instead of virus copies and revised the statement as following: "Considering the immature mice are usually used for evaluating the susceptibility of viruses in the subgenus *Merbecovirus* as reported by previous studies^{12,16}, 5-week-old specific pathogen-free female hDPP4-mice and WT-mice were inoculated intranasally with pangolin-CoV-HKU4-P251T at a volume of 50 µL virus stock with a viral titer of 10⁴ TCID₅₀." (Page 10, Line 210-215)

As suggested by the reviewer, we have assessed the relative expression of sgmRNA in the lung samples of infected mice, and added the results to the revised manuscript as following: "To verify the presence of the replication of the virus, we examined the relative expression of subgenomic mRNA (sgmRNA) in the lung samples. The sgmRNA of the virus was detected only on the first day, and the expression was significantly higher in hDPP4-mice than in WT-mice (Fig. 4b)." (Page 11, Line 231-235)

We appreciated the reviewer's comment on the antigenic data. Pangolin-CoV-HKU4-P251T like other coronaviruses in subgenus *Merbecovirus* is also quickly cleared as shown that sgmRNA in the lung samples was undetected 3 DPI. Although there is no alive virus, the residual viral nucleotides and proteins in the tissues might lead to the positive results of qRT-PCR and IHC. In response to the reviewer's concern, we have added discussions on this issue in the revised manuscript as following: "Either viral loads or antigens are sustained in the lungs of infected hDPP4-mice during the observation period from 1 to 12 DPI, which are comparable to the results observed in the experimental study of Tylonycteris-bat-CoV-HKU4 using hDPP4-transgenic mice¹⁶. While the results of undetectable sgmRNA since 3 DPI in the lung samples indicate that there is no alive virus thereafter. The positive results of qRT-PCR and IHC might be due to the residual viral nucleotides and proteins retained in the tissues." (Page 18, Line 378-384)

In response to the reviewer's inquiry, we have compared the viral genome sequences at different time points, and given the results in the revised manuscript as following: "There was no meaningful non-synonymous mutation identified through the genome sequence analysis of the infected lungs collected at different times after infection (Supplementary Fig. 3d)." (Page 12, Line 246-249)

8) The source of the DPP4 transgenic mice is unclear in the methods of the manuscript.

How is expression driven? Has this mouse resource been reported previously and if not, has hDPP4 expression been evaluated in various tissues of these animals?

Response: We appreciate the reviewer's inquiry, and have refined the description of the source of hDPP4-transgenic mice in Methods of the revised manuscript as following: "hDPP4-transgenic mice and wild-type C57BL/6 mice were used in this study. Five weeks old hDPP4 female mice (Cat. NO. NM-HU-190042) were purchased from Shanghai Model Organisms Center, Inc. (<https://www.modelorg.com/>). The DPP4 targeting construct was designed to insert a DPP4 gene and a FLAG and a SV40 polyA signal into exon3 of the targeted gene. This construct was inserted into the targeted gene via CRISPR/Cas9 system in C57BL/6 mouse background. The donor vector with sgRNA and Cas9 mRNA were microinjected into C57BL/6 fertilized eggs. F0 generation mice positive for homologous recombination were identified by long PCR. The PCR products were further confirmed by sequencing. F0 mice were crossed with C57BL/6 mice to obtain *Dpp4^{+DPP4}* heterozygous mice. Five weeks old wild-type female mice were obtained from Beijing Weitong Lihua Biotechnology Co., Ltd." (Page 32, Line 673-684)

In addition, we performed western blotting assay to test the hDPP4 expression in tissues of the experimental mice, and added the results in the revised manuscript as following: "The western blotting analysis revealed hDPP4 expression in all tissues collected from hDPP4-transgenic mice, while no expression was observed in any tissues from wild-type mice (Supplementary Fig. 3b)." (Page 11, Line 221-223) Accordingly, we described the detection of hDPP4 expression in Methods of the revised manuscript as following: "The hDPP4 expression was detected using western blotting assay in all the collected tissues of both hDPP4-transgenic and wild-type mice." (Page 34, Line 731-734) ""

9) Lines 193-205. PgCoV P251T mice did not lose weight after infection (Fig 5). The DPP4 transgenic animals didn't gain weight like controls, which is a different phenotype. This section needs to be written to accurately reflect the phenotype. Weight loss statements are simply incorrect and inappropriate. It is strange that only macs are increasing. No T cell responses or antibody responses in these animals by day 12?

Response: We appreciate the reviewer's comment on body weight change, and have re-written our statements to avoid confusion in the revised manuscript as following: "The percentages of body weight change for each group were estimated by comparing the group average weight with their initial average weight. The body

weights of hDPP4-mice in the control group maintained about 10% gain since 1 DPI. While the weights of pangolin-CoV-HKU4-P251T-infected hDPP4-mice always lower than those of control groups inoculated with either cell culture supernatant or heat-inactivated virus during the experiment period, and was decreased since 5 DPI (all $P < 0.05$), with a weight loss of up to 4.6% at 9 DPI (Fig. 5a). The body weights of WT-mice infected with pangolin-CoV-HKU4-P251T kept comparable to those of uninfected control WT-mice. Although no obvious clinical sign was observed, the hDPP4-mice infected with pangolin-CoV-HKU4-P251T did not gain weight in contrast to either the uninfected hDPP4-mice or wild-type mice, and even showed body weight loss in the late stage of infection, suggesting that hDPP4-mice had been affected by the virus infection.” (Page 12-13, Line 254-266)

In response to the reviewer’s inquiry about the inflammatory response in lung tissues of the infected mice, we have provided more detailed discussion about the predominate presence of macrophage in the revised manuscript as following: “The progressive infiltration and aggregation of inflammatory cells, mainly macrophages, were observed in the infected lung tissues of hDPP4-transgenic mice. This kind of inflammatory response is something similar to that caused by SARS-CoV-2 infection in hACE2-transgenic mice, from which aggregated macrophages and only a few T lymphocytes or B lymphocytes were detected in the lung²¹ Why only macrophages predominately present in the inflammatory lung tissues of experimental mice after pangolin-CoV-HKU4-P251T or SARS-CoV-2 infection, is unclear and deserves further investigation.” (Page 18, Line 385-392)

10) Discussion.

a. Only no-nsynonomous mutations over 15 passages? Include the RNA seq data for several different passages as reviewers need to be able to examine these data directly as it seems unlikely and needs to be supported with raw read data.

Response: We appreciate the reviewer’s inquiry, and have further analyzed the genome sequences through 15 passages and provided the sequence data in Supplementary Table 3. As our above response to the comment #2, we have added Fig. 1d and described the results in the revised manuscript as following: “We conducted viral sequencing of the 15 passages to assess possible mutation of pangolin-CoV-HKU4-P251T through subcultures (Supplementary Table 3), and found that it had a persistent synonymous substitution at position 2702 (AGC:AGT) of ORF1ab and a consistent non-synonymous substitution of A2P in NS3d protein from the fourth to 15th passage. A non-synonymous substitution leading to A843V in spike protein occurred since the fifth passage (Fig. 1d). These

findings indicate the virus was rather stable during serial passages *in vitro*.” (Page 6, Line 116-124) In addition, we have discussed the mutation in S protein in the revised manuscript as following: “Given that the passage 15 exhibited higher growth rates, the A843V substitution in the S protein might be associated with efficient virus replication, which deserves further investigation.” (Page 15, Line 316-318)

b. The authors argue that the P251 virus may not only use DPP4 in its natural host. Given the recent published literature of ACE2 using Merbecoviruses, they should test whether its using ACE2 in its natural host

Response: We appreciate the reviewer’s valuable suggestions, and have performed supplementary experiments. As our above response to the comment #6, the results have been provided to the revised manuscript as following: “To further investigate whether pangolin-CoV-HKU4-P251T can utilize DPP4 and ACE2 as receptors for infection, we conducted the experiments by observing the virus infectivity in wild type HeLa (HeLa-WT) cells as well as human DPP4-expressing HeLa (HeLa-hDPP4), *Manis javanica* pangolin DPP4-expressing HeLa (HeLa-pDPP4) and *T. pachypus* bat DPP4-expressing HeLa (HeLa-bDPP4), human ACE2-expressing HeLa (HeLa-hACE2), pangolin ACE2-expressing HeLa (HeLa-pACE2) and bat ACE2-expressing HeLa (HeLa-bACE2) cells (Fig. 3a and c). Pangolin-CoV-HKU4-P251T grew well and replicated efficiently in hDPP4-HeLa and pDPP4-HeLa cells, with 21.8 and 64.5-fold increase in viral copies, respectively, in comparison to viral copies in HeLa-WT cells at 48 HPI. The fold increase in viral copies was 5.0 in HeLa-bDPP4 cells, which was significantly lower than those in HeLa-hDPP4 and HeLa-pDPP4 cells (Fig. 3b). In contrast, ACE2-expressing cells did not show any increase in viral load compared to HeLa-WT cells, suggesting that ACE2 may not be an essential receptor for pangolin-CoV-HKU4-P251T (Fig. 3d).” (Page 9, Line 178-191)

c. Discussion throughout. The virus doesn’t cause any body weight loss. It prevents body weight gain. A very different phenotype and it needs to be discussed properly in the discussion. As currently written, the statement is inaccurate.

Response: We appreciate the reviewer’s comment on body weight change. As our above response to the comment #9, we have re-written our statements to avoid confusion in the revised manuscript as following: “The percentages of body weight

change for each group were estimated by comparing the group average weight with their initial average weight. The body weights of hDPP4-mice in the control group maintained about 10% gain since 1 DPI. While the weights of pangolin-CoV-HKU4-P251T-infected hDPP4-mice always lower than those of control groups inoculated with either cell culture supernatant or heat-inactivated virus during the experiment period, and was decreased since 5 DPI (all $P < 0.05$), with a weight loss of up to 4.6% at 9 DPI (Fig. 5a). The body weights of WT-mice infected with pangolin-CoV-HKU4-P251T kept comparable to those of uninfected control WT-mice. Although no obvious clinical sign was observed, the hDPP4-mice infected with pangolin-CoV-HKU4-P251T did not gain weight in contrast to either the uninfected hDPP4-mice or wild-type mice, and even showed body weight loss in the late stage of infection, suggesting that hDPP4-mice had been affected by the virus infection.” (Page 12-13, Line 254-266)

We also revised the statement about the weight change in Discussion as following: “Therefore, we select young mice for the *in vivo* experiments, and find that pangolin-CoV-HKU4-P251T shows its infectivity and pathogenicity in hDPP4-mice, characterized by the affected weight gain and interstitial pneumonia after intranasal inoculation.” (Page 18, Line 375-378)

d. Line 287 “may be” is a more appropriate verb usage here, as the authors haven’t demonstrated virus growth in primary human airway or intestinal cells. A pre-requirement is showing such data, alternatively the limitation of the study should be directly acknowledged. Primary and continuous cells in culture are different, as are proteases, which may play a role in cross species transmission and replication.

Response: As suggested by the reviewer, we have revised the sentences to avoid the overstated conclusions as following: “The efficient replication of pangolin-CoV-HKU4-P251T in the three human cell lines suggests the pangolin-borne *Merbecovirus* may infect humans, once it spills over in an adaptable circumstance.” (Page 17, Line 348-351)

In addition, we have acknowledged the limitation in Discussion as following: “One limitation of this study is that the infectivity and replication in primary human cells are unknown, although pangolin-CoV-HKU4-P251T grows well in multiple continuous human cell lines, Therefore, we cannot definitely conclude the cross-species transmission potential of the virus at this moment, because the cultures of primary and continuous cells contain different proteases, which may play a role in susceptibility to the virus. A further evaluation of virus growth in primary human airway or intestinal cells will provide more direct evidence of spill-

over transmission possibility of the pangolin-associated virus.” (Page 19, Line 399-406)

e. Lines 308+ significant body weight loss needs to be changed to prevent significant increases in body weight gain after infection (throughout discussion).

Response: We are very grateful for the reminding, and have modified the statements about weight change wherever necessary throughout the manuscript.

f. Line 326 or so. A recent publication in Cell describes a Merbecovirus that uses hDPP4 receptors and grows in mice. It needs to be acknowledged and discussed in the discussion. What are the similarities and differences in the two studies.

Response: We appreciate the reviewer’s comments, and have revised the discussion regarding the recently published MjHKU4r-CoV-1 as following: “Isolation of the coronaviruses is very important to do experiments for better understanding their cross-species transmissibility and potential for emergence as human pathogens. In the current study, we successfully isolate a virus in subgenus *Merbecovirus*, from a Malayan pangolin. Pangolin-CoV-HKU4-P251T infects hDPP4-transgenic mice leading to prominent inflammatory damages of lungs. Our findings provide further evidence similar to the results of a recent report using closely related isolate of MjHKU4r-CoV-1¹². Both studies suggest that the pangolin-borne coronaviruses in subgenus *Merbecovirus* may be capable to spillover from pangolins to humans by using DPP4 receptors.” (Page 19-20, Line 409-421)

Responses to Reviewer #2:

1. The manuscript is not as well written as articles traditionally published in this publication. It is at times difficult to follow, which detracts from the science. The authors need to improve the language with regard to grammar and word usage, as well as general flow.

Response: We appreciate the reviewer’s concern. As suggested by the reviewer, the

manuscript has been reviewed and polished by a senior researcher with fluent English language.

2. Several passages that are currently in the Results section should go into the Discussion section (e.g. line 134-138, line 109-110)

Response: As the suggestion by the reviewer, we have removed the original line 134-138 from Results, and integrated the relevant statements in the Discussion section as following : “While the sequence and binding analyses have indicated that the spike protein from bat HKU4 could recognize both hDPP4 and bDPP4 as its receptors^{14,15}, the experiments using live *Tylonycteris*-bat-CoV-HKU4 indicate that it can only use hDPP4 and dromedary camel-DPP4 (dcDPP4) as receptors, but not infect bDPP4-expressing cells or primary kidney and lung cells of *T. pachypus* bats, from which the virus was isolated¹⁶. Our findings together with those from the related study on *Tylonycteris*-bat-CoV-HKU4 isolate suggest that bat DPP4 may not be essential for *Tylonycteris*-Bat-CoV-HKU4 in its natural host, nor for its closely related merbecoviruses, such as pangolin-CoV-HKU4-P251T, from other animal hosts.” (Page 17, Line 362-370).

We have moved the original line 109-110 from Results to Discussion section with modification as following: “Given that the passage 15 exhibited higher growth rates, the A843V substitution in the S protein might be associated with efficient virus replication, which deserves further investigation.” (Page 15, Line 316-318)

3. What is the rationale for the claim in Line 109- 110 that ‘the A843V substitution in spike protein might be associated with viral activity’? - This should also be in the Discussion, including their analysis in reaching the conclusion, and other possible explanations, if any. [For example, could this be an artifact?]

Response: We appreciate the reviewer’s valuable comments. As suggested also by another reviewer, we have conducted an additional experiment to evaluate the kinetics of virus replication of original isolate and the adapted virus with A843V substitution in S protein. We have added the results in the revised manuscript as following: “Furthermore, we compared the kinetics of virus replication between the original isolate (passage 1) and *in vitro* adapted strain (passage 15) that used for the experiments in this study. The adapted viruses grew more efficiently in the initial stage reaching peak at 48 HPI, and slowed down thereafter. While the titer

of the originally isolated virus was gradually increased and up to the comparable level with the adapted virus at 144 HPI (Fig. 2f).” (Page 8, Line 159-164) As mentioned above, we have modified the discussion as following: “Given that the passage 15 exhibited higher growth rates, the A843V substitution in the S protein might be associated with efficient virus replication, which deserves further investigation.” (Page 15, Line 316-318)

4. Line 123

Here, the authors conclude that ‘the signal intensity at 48 HPI in each cell line was in accordance with the viral load result detected by qRT-PCR’. – To compare staining in cells to a PCR signal, the authors need to quantify the signal strength of the intensity of the FISH signal in the individual samples.

Response: We appreciate the reviewer’s concern. Because the FISH signal cannot be easily quantified due to lack of the equipment in our laboratory, we have modified the inappropriate statement on FISH results in the revised manuscript as following: “We conducted FISH using specific probes, and the viral signals were visualized in Huh7, Caco-2, Calu-3 and BEAS-2B cells (Fig. 2b), which confirmed the infectivity of pangolin-CoV-HKU4-P251T in multiple human cells.” (Page 7, Line 138-142)

5. What was the rationale for using immunologically immature mice? The authors should address this decision, as the age of the mice could have influenced the results shown. Specifically, Coronavirus infection has already been shown to be age-dependent. The authors should consider an additional experiment in aged mice; this model might develop pathology. At least, this possibility should be addressed in the Discussion.

Response: We appreciate the reviewer’s inquiry and suggestion, and have provided the rational with the references as well as discussed the age-dependent susceptibility of mice to coronaviruses in different subgenera in the revised manuscript as following: “Considering the immature mice are usually used for evaluating the susceptibility of viruses in the subgenus *Merbecovirus* as reported by previous studies^{12,16}, 5-week-old specific pathogen-free female hDPP4-mice and WT-mice were inoculated intranasally with pangolin-CoV-HKU4-P251T at a volume of 50 μ L virus stock with a viral titer of 10^4 TCID₅₀.” (Page 10, Line 210-215) “Coronavirus infections have already been shown to be age-dependent: The aged mice tend to be more susceptible to the viruses in subgenus *Sarbecovirus* such

as SARS-CoV and SARS-CoV-2^{36,37}, while the experimental studies to evaluate the susceptibility of viruses in the subgenus *Merbecovirus* usually use immature mice^{12,16}. Therefore, we select young mice for the *in vivo* experiments, and find that pangolin-CoV-HKU4-P251T shows its infectivity and pathogenicity in hDPP4-mice, characterized by the affected weight gain and interstitial pneumonia after intranasal inoculation.” (Page 17-18, Line 371-378)

Responses to Reviewer #4:

The authors identified an HKU4 related coronavirus from a Malayan pangolin. They minced intestinal tissue from the pangolin and inoculated Vero and other cell lines. They detect viral genome signal and FISH staining for orf1ab in some cell types that they describe as efficient replication. However, they do not describe any cytopathic effect and the increase in genome copies is less than 100-fold. Infection of hDPP4 transgenic mice is not convincing. There is no evidence of an increase in genome replication in the lungs of mice; there is a slower weight gain but no weight loss and the histologic changes are not convincing evidence of viral replication. The authors should consider inoculating a control group of animals with heat or UV inactivated virus.

Response: We are very grateful for the valuable comments and suggestions. The original manuscript did not describe the viral cytopathic effect (CPE) because the lesions caused by the virus in Vero 81 cells are so weak that it is difficult to determine. With some experimentation, we found that the virus caused CPE in Huh7 cells, so we added the result to the main text to demonstrate the activity of the virus as following: “Notably, obvious cytopathic effect (CPE) was visualized in the pangolin-CoV-HKU4-P251T-infected Huh7 cells (Fig. 2d), while CPE was subtle in either infected Vero 81 or Caco-2 and Calu-3 cells.” (Page 8, Line 154-156)

We have revised the statement of animal experiments to make our conclusion more rigorous as following: “The body weights of hDPP4-mice in the control group maintained about 10% gain since 1 DPI. While the weights of pangolin-CoV-HKU4-P251T-infected hDPP4-mice always lower than those of control groups inoculated with either cell culture supernatant or heat-inactivated virus during the experiment period, and was decreased since 5 DPI (all $P < 0.05$), with a weight loss of up to 4.6% at 9 DPI (Fig. 5a). The body weights of WT-mice infected with pangolin-CoV-HKU4-P251T kept comparable to those of uninfected control

WT-mice. Although no obvious clinical sign was observed, the hDPP4-mice infected with pangolin-CoV-HKU4-P251T did not gain weight in contrast to either the uninfected hDPP4-mice or wild-type mice, and even showed body weight loss in the late stage of infection, suggesting that hDPP4-mice had been affected by the virus infection.” (Page 12-13, Line 255-266) “Therefore, we select young mice for the *in vivo* experiments, and find that pangolin-CoV-HKU4-P251T shows its infectivity and pathogenicity in hDPP4-mice, characterized by the affected weight gain and interstitial pneumonia after intranasal inoculation.” (Page 18, Line 375-378)

As suggested by the reviewer, we have added another control group inoculated with heat-inactivated virus, and provided the results of this group into appropriate places in the text of the revised manuscript (Page 11, Line 215-216) (Page 11, Line 230-231) (Page 12-13, Line 257-266) (Page 34, Line 724-725).

Minor comments:

1. Line 52: change the word denominated to named

Response: Done as suggested. (Page 3, Line 52)

2. Line 54: in silico

Response: Done as suggested. (Page 3, Line 56)

REVIEWER COMMENTS

Reviewer #2 (Remarks to the Author):

The authors have addressed my comments to my full satisfaction.

Reviewer #4 (Remarks to the Author):

The authors have revised the paper quite extensively based on the reviewer's comments especially the inclusion of the other pangolin HKU4-related virus. They have included Huh-7 cells in which they detected cytopathic effect and they now report modest titres of infectious virus. In hDPP4 transgenic mice, they have now included an inactivated virus group but interestingly this group of mice also did not gain weight. Subgenomic RNA was detected only on day 1. Thus, the evidence of productive infection in hDPP4 mice is still weak. Evidence of seroconversion in the mice would strengthen the inference that the mice were infected. Using older mice may also give different results.

Statements (eg line 307) about prolonged pangolin-CoV-HKU4-P251T infection should be more precisely described as prolonged detection of viral RNA.

Minor comments:

Line 247: what are the authors trying to convey with the term 'meaningful' about non-synonymous mutations?

Line 382: change alive virus to live or infectious virus.

Responses to Reviewer #4:

The authors have revised the paper quite extensively based on the reviewer's comments especially the inclusion of the other pangolin HKU4-related virus. They have included Huh-7 cells in which they detected cytopathic effect and they now report modest titres of infectious virus. In hDPP4 transgenic mice, they have now included an inactivated virus group but interestingly this group of mice also did not gain weight. Subgenomic RNA was detected only on day 1. Thus, the evidence of productive infection in hDPP4 mice is still weak. Evidence of seroconversion in the mice would strengthen the inference that the mice were infected. Using older mice may also give different results.

Response: We appreciate the reviewer's valuable suggestions, and have performed supplementary experiments to evaluate the seroconversion of mice. The results have been added to the revised manuscript as following: "To evaluate the seroconversion of virus-infected mice, serum was collected from all mice, and the serum neutralizing antibody assay against pangolin-CoV-HKU4-P251T was conducted. Neutralizing antibodies were developed in one-third of virus-infected hDPP4-mice, whereas no neutralizing antibody was detected in the serum of the virus-infected WT-mice and any control mice (Supplementary Table 4)." (Page 11, Line 217-222) We also discuss this results in Discussion section as following: "Moreover, we also found that virus-infected hDPP4-mice developed measurable neutralizing antibodies distinct from the WT-mice and any controls, which further demonstrated the productive infection of pangolin-CoV-HKU4-P251T in hDPP4-mice." (Page 18, Line 369-372) Relevant experimental details can be seen in the Methods section as following: "Both hDPP4- and WT-mice serum were collected at each time point after virus infection to measure serum neutralizing antibody titers. Huh7 cells were seeded in 96-well plates and incubated overnight at 37°C under 5% CO₂ to allow formation of cell monolayers. All individual serum were heat-inactivated for 30 min at 56 °C, serially diluted (2-fold dilutions starting at 1:10), and were assayed against 100 TCID₅₀ of virus in 96-well plates (1:1 mixtures). The control mice serum and cell culture medium served as negative control and the pure virus as positive control. The plates were incubated at 37 °C for five days, and cytopathic effects were observed. Serum neutralizing antibody

titers were expressed as 50% neutralizing antibody titers (NT50).” (Page 37-36, Line 793-803)

We also appreciate the reviewer for suggesting using older mice and agree that the older mice may exhibit different phenotypes. Regrettably, however, limited experimental conditions (e.g., ABSL-3 laboratory) and difficult access to older hDPP4-mice preclude us from performing this virus infection experiment in a short period. All in all, we sincerely thank the reviewer for the valuable suggestion which sheds new light on our future studies.

Statements (eg line 307) about prolonged pangolin-CoV-HKU4-P251T infection should be more precisely described as prolonged detection of viral RNA.

Response: We appreciate the reviewer’s reminder, and have refined the description as following: “The persistent increases in main cytokines observed in lung tissues of the hDPP4-mice imply the prolonged persistence of viral RNA, which deserves further investigation.” (Page 14, Line 293-295)

Minor comments:

Line 247: what are the authors trying to convey with the term ‘meaningful’ about non-synonymous mutations?

Response: We appreciate the reviewer’s inquiry, and have detailed the statement as following: “There was no meaningful non-synonymous mutation associated with virus host-adaptability and pathogenicity identified through the genome sequence analysis of the infected lungs collected at different times after infection (Supplementary Fig. 3d).” (Page 11, Line 233-236)

Line 382: change alive virus to live or infectious virus.

Response: Done as suggested. (Page 17, Line 367)

REVIEWERS' COMMENTS

Reviewer #4 (Remarks to the Author):

The authors have included data from a neutralising antibody assay in response to my suggestion that serological evidence of infection would strengthen the inference that the mice were infected. Examining sera earlier than 7 days post infection is not likely to be informative because neutralising antibody responses are unlikely to be elicited so early. Two of 3 mice on day 9 post infection had detectable neutralising antibodies but 0 of 3 mice at day 12 post infection had detectable neutralising antibodies.

Thus, serological data are also not convincing and the conclusion that mice were infected remains weak.

it would be better to infect some mice and test sera at days 14, 21 and 28 post infection.

Responses to Reviewer #4:

The authors have included data from a neutralising antibody assay in response to my suggestion that serological evidence of infection would strengthen the inference that the mice were infected. Examining sera earlier than 7 days post infection is not likely to be informative because neutralising antibody responses are unlikely to be elicited so early. Two of 3 mice on day 9 post infection had detectable neutralising antibodies but 0 of 3 mice at day 12 post infection had detectable neutralising antibodies.

Thus, serological data are also not convincing and the conclusion that mice were infected remains weak.

It would be better to infect some mice and test sera at days 14, 21 and 28 post infection.

Response:

We appreciate the great suggestions of the reviewer, especially the neutralizing antibody testing of sera on days 14, 21, and 28 post-infection. Due to access constraints in the BSL-3 laboratory, we are unfortunately unable to perform such new experiments in the short term. We acknowledge that productive infection cannot be adequately substantiated, because of the limited time for testing the neutralizing antibody. Therefore, as suggested by the editor, we have toned down the conclusions regarding infection of transgenic mice, and revised the title and abstract of the manuscript. Now, the title has been changed to “Isolation and characterization of a pangolin-borne HKU4-related coronavirus that potentially infects human-DPP4-transgenic mice”, and the corresponding part of the abstract has been changed to “After intranasal inoculation with pangolin-CoV-HKU4-P251, hDPP4-transgenic female mice are likely infected, showing persistent viral RNA copy numbers in the lungs.” (Page 2, Line 30-32)